J Physiol 603.19 (2025) pp 5777–5804

# Hypoxia-inducible factor-2 stabilization is not sufficient to induce erythropoietin production in deeper medullary fibroblasts

Bettina K. M. Firmke[1] (ID), Michaela A. A. Fuchs[1] (ID), Lena M. Süß[2], Anna-Lena Forst[2] (ID), Armin Kurtz[1] and Katharina A.-E. Broeker[1] (ID)

[1] Physiology I; Institute of Physiology, University of Regensburg, Regensburg, Germany
[2] Medical Cell Biology, Institute of Physiology, University of Regensburg, Regensburg, Germany

Handling Editors: Kim Barrett & Morag Mansley

The peer review history is available in the Supporting Information section of this article (https://doi.org/10.1113/JP288798#support-information-section).

**Abstract figure legend** Renal erythropoietin (EPO) production was analysed in anaemic and PHD-inhibitor-treated wild-type mice, and mice with genetic fibroblast-specific hypoxia-inducible factor (HIF)-2 stabilization. We found, that PHD inhibition activated HIF-2 signalling in interstitial fibroblasts of all kidney zones as observed by the induction of the HIF-2 target genes *Adm* (adrenomedullin) and *Rgs4* (regulator of G protein signalling 4). However, the HIF-2 stabilization was not sufficient to induce EPO expression in deeper medullary fibroblasts. Moreover, two different subsets of interstitial fibroblasts were detected evenly distributed across all kidney zones – PHD2$^+$ (blue cells) and PHD2/3$^+$ fibroblasts (purple cells; about 35%). Solely PHD2$^+$ fibroblasts were the predominant EPO producers under anaemic conditions in the kidney cortex (red cells), while PHD3 in general delayed HIF-2 stabilization and induction of HIF-2 target genes in PHD2/PHD3$^+$ fibroblasts. Overall, our data suggest the existence of additional regulatory mechanisms that control EPO expression in deeper medullary fibroblasts. Created with BioRender.com®.

**Abstract**  Under hypoxaemic conditions, cortical fibroblasts primarily produce erythropoietin (EPO). However, we have previously shown that most interstitial fibroblasts positive for platelet-derived growth factor receptor $\beta$ (PDGFR-$\beta$) in all kidney zones are also able to produce EPO. Therefore, we wondered if either the physiological stimuli might not be sufficient to stabilize the hypoxia-inducible factor (HIF)-2 in medullary fibroblasts or if different expression patterns or functions of the HIF-regulating prolyl-4-hydroxylases (PHD) 2 and 3 might explain the restrictive EPO cell recruitment. This study shows that although HIF-2 can be clearly stabilized in deeper medullary fibroblasts by pharmacological PHD-inhibition, this is not sufficient to induce EPO in these cells. In contrast, genetic stabilization of HIF-2 by cell-specific deletion of either PHD2 or PHD2 and PHD3 in mice resulted in EPO production in all kidney zones. EPO induction in PHD2/3-KO mice was twice as high as in PHD2-KOs. PHD3 deletion slightly increased basal EPO expression. Accordingly, in contrast to PHD2, PHD3 expression was only detected in a subset of interstitial fibroblasts, without zonal accumulation or hypoxaemic upregulation. Exposure of PHD3-deficient mice to a hypoxaemic stimulus resulted in significantly higher EPO levels compared to controls, with EPO induction restricted to the cortex. Overall, our data suggest the existence of additional regulatory mechanisms beyond the HIF-2 signalling pathway that control EPO expression in deeper medullary fibroblasts. Furthermore, they identify PHD3 as an attenuating factor that delays EPO induction in a subset of cortical PDGFR-$\beta^+$ cells, but its expression pattern is not the determining factor responsible for the cortical restriction of EPO.

(Received 26 February 2025; accepted after revision 31 July 2025; first published online 21 August 2025)

**Corresponding author** K. A.-E. Broeker: Institute of Physiology, University of Regensburg, Universitätsstra$\beta$e 31, D-93053 Regensburg, Germany.    Email: Katharina.Broeker@ur.de

**Key points**

- Pharmacological prolyl-4-hydroxylase (PHD) inhibition activates hypoxia-inducible factor (HIF)-2 signalling in interstitial fibroblasts from all renal zones.
- HIF-2 stabilization is not sufficient to induce erythropoietin (EPO) expression in deeper medullary fibroblasts, although they are in principle capable of producing EPO.
- There are two subsets of interstitial fibroblasts, PHD2$^+$ and PHD2/PHD3$^+$ fibroblasts, that are evenly distributed throughout the kidneys, thus also not determining the restrictive cortical induction of EPO under hypoxaemic and pharmacological conditions.
- Solely PHD2$^+$ fibroblasts are the predominant EPO producers in the renal cortex, while PHD3 is an attenuating factor that delays EPO induction in PHD2/PHD3$^+$ fibroblasts. HIF2-induced upregulation of PHD isoforms is not detectable in interstitial fibroblasts.
- Overall, our data suggest the existence of additional regulatory mechanisms, in addition to the HIF-2 signalling pathway, that control EPO expression in deeper medullary fibroblasts.

## Introduction

Fibroblast and pericyte-like cells positive for platelet-derived growth factor receptor $\beta$ (PDGFR-$\beta$) are a numerous cell population in the kidney that are often associated with their status as precursors of myofibroblasts in fibrotic diseases. However, they also fulfil important functions in the healthy kidney, for example,

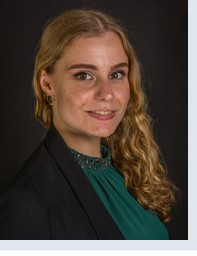

**Bettina Firmke** studied biochemistry and molecular science at the University of Bayreuth. She is now a PhD student in Dr Katharina Broeker's group at the Institute of Physiology at the University of Regensburg. She is interested in elucidating the endocrine plasticity of renal interstitial cells in healthy and fibrotic kidneys. In the future, she will focus her research on characterizing these interstitial cells and unravelling their physiological functions in health and disease, such as in cases of chronic kidney disease.

as producers of endocrine factors such as erythropoietin, renin, prostaglandins or preproenkephalin A (Penk), the precursor of endogenous opioid-like factors. The erythropoietin (EPO)- or renin-producing fibroblasts are usually found in the cortex and the outer stripe of the outer medulla (Broeker et al., 2021; Eckardt et al., 1993; Maxwell et al., 1993), while prostaglandins and Penk tend to be produced by fibroblasts/pericytes in the deeper medullary zones of healthy kidneys (Fuchs, Schrankl, Leupold, et al., 2022; Fuchs, Schrankl, Wagner, et al., 2022; Steinert et al., 2009). Overall, it seems that renal fibroblasts are divided into at least two functional subpopulations. Under certain conditions, however, the zonal boundaries and functional organization seem to become blurred. We and others could show that, for example, in anaemia or after pharmacological EPO induction, cortical fibroblasts can express both EPO and renin simultaneously (Broeker et al., 2021; Miyauchi et al., 2021). More interestingly, we found that genetic activation of the hypoxia-inducible signalling pathway, which regulates EPO expression, led to EPO production in most PDGFR-$\beta^+$ fibroblasts in all renal zones, despite the fact that under physiological conditions EPO production is restricted to the cortex and the cortico-medullary border (Broeker et al., 2020; Eckardt et al., 1993; Gerl et al., 2016).

It has long been known that EPO is not regulated by increasing its expression per cell, but by recruiting additional interstitial fibroblasts in which EPO production is transcriptionally switched on. The main transcription factor regulating EPO is hypoxia-inducible factor (HIF)-2$\alpha$, whose stabilization is dependent on oxygen availability (Koury et al., 1989; Paliege et al., 2010; Wenger & Hoogewijs, 2010). Functional HIFs consist of two subunits, an $\alpha$- and a $\beta$-subunit. Under normoxia, the HIF-$\alpha$ subunit is hydroxylated by prolyl-4-hydroxylases (PHDs), ubiquitinylated by the von-Hippel–Lindau protein (Vhl) and then proteasomally degraded. However, under hypoxic conditions, HIF-$\alpha$ subunits can no longer be hydroxylated and labelled for proteasomal degradation. Instead, HIF-$\alpha$ translocates to the nucleus where it dimerizes with the HIF-1$\beta$ subunit and then activates the transcription of target genes, such as EPO, in complex with other cofactors (Haase, 2006). Three isoforms of PHDs are known – PHD1, PHD2, and PHD3 (Kaelin & Ratcliffe, 2008). In the context of EPO regulation, PHD2 is considered to be the most important isoform, while PHD3 is reported to have a more modulating influence on HIF stabilization in general and also on EPO production (Broeker et al., 2021; Kobayashi et al., 2016; Minamishima et al., 2008, 2009). In the clinic pharmacological inhibition of PHD activity is used to overcome insufficient EPO production, most commonly seen in situations of renal disease (Haase, 2021).

Looking at the recruitment of renal EPO cells resulting from hypoxaemic stimuli like anaemia, low tissue oxygenation or carbon monoxide exposure, a distinct pattern can be observed depending on the strength of the hypoxaemic stimuli (Eckardt et al., 1993). Recruitment of additional EPO cells starts in a cluster-like fashion along the corticomedullary border and spreads outwards towards the cortex surface. Interestingly, medullary fibroblasts are not recruited for EPO production under physiological hypoxaemic conditions (Broeker et al., 2020; Dahl et al., 2022; Eckardt et al., 1993; Maxwell et al., 1997). This is particularly noteworthy because measurements of renal tissue oxygenation show that the renal medulla is already more hypoxic than the cortex under normoxic conditions (Edwards & Kurtcuoglu, 2022). Notably, even after treatment with a single dose of a PHD inhibitor, which induces EPO independently of the tissue oxygenation, EPO cell recruitment mimics the hypoxaemic expression pattern and is restricted to the cortex and corticomedullary border (Dahl et al., 2022; Fuchs et al., 2021; Kobayashi et al., 2022). In contrast, we could previously show, that when HIF-2 was genetically stabilized in interstitial PDGFR-$\beta^+$ fibroblasts by deleting Vhl, EPO expression was induced in most interstitial fibroblasts in all kidney zones, including the papilla (Gerl et al., 2016).

These findings raise the question of why medullary fibroblasts do not produce EPO – neither under hypoxaemic conditions nor due to PHD inhibition – although they are competent to do so?

Various factors are conceivable as an explanation for the lack of EPO expression in medullary fibroblasts: in contrast to the genetic stimulus with its lack of the Vhl protein, the physiological hypoxaemic stimuli or a single pharmacological stimulus may not be sufficient to stabilize HIF-2 in deeper medullary fibroblasts, either because of differences in their HIF-2$\alpha$ expression levels compared to cortical fibroblasts, or because they are less susceptible to HIF-2 stabilization at the protein level. Alternatively, medullary fibroblasts may differ from cortical fibroblasts in their PHD2 and PHD3 expression patterns and therefore have higher thresholds for HIF-2 stabilization. This would be consistent with our findings regarding the metaplastic transformation of juxtaglomerular renin-producing cells into EPO-producing cells. Here, renin-cell-specific deletion of PHD2 was not sufficient to stabilize HIF-2 and induce EPO. Only co-deletion of PHD2 and PHD3 was able to induce a phenotypic transformation of the renin cells into EPO cells (Broeker et al., 2021).

To address these different possibilities, we examined interstitial fibroblasts for zonal differences in HIF-2$\alpha$ mRNA expression, HIF-2 stabilization and HIF-2 pathway activation under normoxic and hypoxaemic conditions as well as after prolyl-4-hydroxylase inhibitor (PHDi) treatment and genetic HIF-2 activation.

**Table 1. Primer sequences used for genotyping**

| Genotype | Sequence (5′ to 3′), fwd | Sequence (5′ to 3′), rev | Sequence (5′ to 3′), rev2 |
|---|---|---|---|
| PDGFR-$\beta^{CreERT2}$ | GAACTGTCACCGGGAGGA | AGGCAAATTTTGGTGTACGG | |
| PHD2 flox | CGCATCTTCCATCTCCATTT | CTCACTGACCTACGCCGTGT | GGCAGTGATAACAGGTGCAA |
| PHD3 flox | ATGGCCGCTGTATCACCTGTAT | CCACGTTAACTCTAGAGCCACTGA | |
| Vhl flox | CTAGGCACCGAGCTTAGAGGTTTGC | CTGACTTCCACTGATGCTTGTCACAG | |

In addition, we aimed to further elucidate the functional role of PHD isoforms 2 and 3 in the regulation of renal EPO expression. To this end, interstitial fibroblasts were analysed for possible zonal differences in PHD2 and PHD3 expression patterns and for PHD isoform co-expression with EPO under different conditions. Moreover, mouse models with PDGFR-$\beta^+$ cell-specific deletion of PHD2, PHD3 or PHD2/PHD3 were used to investigate the extent and distribution of EPO induction after deletion of the respective PHD isoform.

## Methods

### Ethical approval

All animal experiments were conducted in accordance with Directive 2010/63/EU of the European Parliament and of the Council on the protection of animals used for scientific purposes. The experiments also comply with the animal ethics checklist of this journal (Grundy, 2015). All experiments were approved by the local ethics committee (Regierung von Unterfranken; 55.2 DMS 2532-2-142; RUF-55.2.2-2532-2-1754; RUF-55.2.2-2532-2-2173) and comply with the guidelines of the animal research centre of the University of Regensburg.

### Animals

All mice were kept in controlled conditions: 12:12 h light–dark cycle, controlled temperature levels (22 $\pm$ 2°C) and humidity (55 $\pm$ 10%). Animals were fed a standard rodent chow (0.6% NaCl; Ssniff, Soest, Germany) with free access to autoclaved tap water.

To specifically target PHD or Vhl expression in PDGFR-$\beta^+$ cells, mice expressing a tamoxifen-inducible Cre recombinase under the PDGFR-$\beta$ promotor were used (PDGFR-$\beta^{CreERT2/+}$) (Gerl et al., 2016).

PDGFR-$\beta^{CreERT2/+}$ PHD2$^{ff}$ (PHD2-KO), PDGFR-$\beta^{CreERT2/+}$ PHD3$^{ff}$ (PHD3-KO), PDGFR-$\beta^{CreERT2/+}$ PHD2$^{ff}$ PHD3$^{ff}$ (PHD2/PHD3-KO) and PDGFR-$\beta^{CreERT2/+}$ Vhl$^{ff}$ (Vhl-KO) mice were generated by crossbreeding PDGFR-$\beta^{CreERT2/+}$ mice and mice with loxP-flanked PHD2 (Franke et al., 2013; Singh et al., 2013) and/or PHD3 (Takeda et al., 2006) or Vhl alleles (JAX

stock #012933, The Jackson Laboratory, Bar Harbor, ME) (Haase et al., 2001).

Genotyping was performed using the primers listed in Table 1. Littermates negative for the Cre construct were used as controls. Male and female mice were included in similar numbers in each group. PDGFR-$\beta^{CreERT2}$ activity was induced at an age of 8–12 weeks by feeding the mice a tamoxifen diet (400 mg tamoxifen citrate/kg; A115T00404; Ssniff) for 21 days. After, the mice were fed standard rodent chow for 21 days.

Anaemia was induced by repeated phlebotomy from the facial vein at the lateral side of the cheek. The facial vein was punctured with a thin sterile cannula (22G) and the blood was collected in ammonium heparin-coated capillaries (KABE Labortechnik GmbH, Nümbrecht-Elsenroth, Germany) to determine haematocrit values, plasma EPO concentrations and blood volume. To minimize negative effects on the cardiovascular system, the mice were injected with prewarmed isotonic saline solution (37°C) to replace the lost volume. The mice were closely monitored to ensure their well-being. Four hours after haematocrit values of 25 $\pm$ 2% were reached, kidneys were removed for analysis. To pharmacologically stabilize HIF-2, the PHD inhibitor roxadustat was orally administered to wild-type mice. For this purpose, a ready-to-use solution (10 mg/ml in 0.5 M Tris–HCl buffer, pH 9.0) was prepared from Evrenzo tablets (Astellas Pharma, Munich, Germany) according to pharmaceutical standards. The experimental groups were treated with increasing numbers of doses of roxadustat (each dose equivalent to 50 mg/kg body weight), administered orally at 90-min intervals. The series included 1$\times$, 3$\times$, 6$\times$, 8$\times$ and 10$\times$ roxadustat. The number of doses administered for each experiment/treatment group is noted in the results section. Wild-type anaemia and PHDi-treated experimental groups consisted of males and females in similar numbers (age 12–18 weeks; genetic background C57/Bl6J). All animals were anaesthetized with ketamine/xylazine (ketamine, 100 mg/kg bodyweight; xylazine 10 mg/kg bodyweight, intraperitoneally) for final organ removal. Blood was collected directly from the abdominal artery and the animals died by exsanguination under anaesthesia. After, left kidneys were removed and snap frozen in liquid nitrogen for RNA quantification.

**Table 2. Primer sequences used for qPCR**

| Gene | Sequence (5′ to 3′), fwd | Sequence (5′ to 3′), rev | Product size (bp) |
|---|---|---|---|
| *Rpl32* | TTAAGCGAAACTGGCGGAAAC | TTGTTGCTCCCATAACCGATG | 100 |
| *Actb* | CCCTAGGCACCAGGGTGTG | GCTGGGGTGTTGAAGGTCTC | 282 |
| *Epo* | AATGGAGGTGGAAGAACAGG | ACCCGAAGCAGTGAAGTGA | 174 |
| *Adm* | GACTCGCTGATGAGACGACA | GAACCCTGGTTCATGCTCTG | 145 |
| *Rgs4* | AAT AGA AAC CAC CGC GGC TC | GAA AGC TGC CAG TCC ACA TT | 292 |
| *Hif-1α* | GGGCATGGTAAAAGAAAGTCCCAGT | GCGACACCATCATCTCTCTGGATT | 858 |
| *Epas-1* (HIF-2α) | AATGACAGCTGACAAGGAG | GAGTGAAGTCAAAGATGCTGTGTC | 407 |
| *Egln1* (PHD2) | CTGTGGAACAGCCCTTTTTG | CGAGTCTCTCTGCGAATCCT | 60 |
| *Egln3* (PHD3) | CGTGGAGCCCATTTTTGACA | AGTACCAGACAGTCATAGCGTA | 107 |

Abbreviations: *Rpl32*, ribosomal protein L32; *Actb*, β-actin; *Epo*, erythropoietin; *Adm*, adrenomedullin; *Rgs4*, regulator of G protein signalling 4; *Hif-1α*, hypoxia-inducible transcription factor 1α; *Epas-1*, endothelial PAS domain protein 1; *Egln1/3*, Egl nine homolog 1/3.

Right kidneys were perfusion-fixed for RNAscope or immunohistochemical analysis.

## Determination of haematocrit values and plasma EPO concentrations

Blood samples were collected from the facial vein or the abdominal artery into ammonium heparin-coated capillary tubes (KABE Labortechnik GmbH). After centrifugation (4 min, 12,879 *g*, RT) haematocrit values were determined, and plasma samples were stored at −80°C before use. Plasma EPO concentrations were determined using the Quantikine Mouse EPO ELISA kit (R&D Systems, Wiesbaden, Germany) according to the manufacturer's protocol.

## Determination of mRNA expression by real-time PCR

Total RNA was isolated from kidneys by acid guanidinium thiocyanate–phenol–chloroform extraction as described previously (Chomczynski & Sacchi, 1987) and quantified by photometer. Moloney murine leukaemia virus reverse transcriptase (Thermo Fisher Scientific, Waltham, MA, USA) and 1 μg of the isolated RNA were used for cDNA synthesis. To quantify mRNA abundance, real-time PCR was performed using the LightCycler SYBR Green I Master Kit and the LightCycler 96 SW instrument (Roche Diagnostics, Mannheim, Germany). Transcript levels were normalized to the housekeeping proteins ribosomal protein L32 (*Rpl32*) or β-Actin (*Actb*). Primers (Eurofins, Munich, Germany) used are listed in Table 2.

## *In situ* hybridization via RNAscope®

For RNAscope analysis kidneys were perfusion-fixed with 40 ml of 10% neutral buffered formalin solution. The fixed tissue was dehydrated, embedded in paraffin, and cut into 5 μm sections with a microtome as described previously (Broeker et al., 2021). Target mRNAs were hybridized and detected with the RNAscope® Multiplex Fluorescent v2 kit (Advanced Cell Diagnostics ACD, Hayward, CA, USA) according to the manufacturer's protocol (Wang et al., 2012). The TSA Vivid dyes 570 and 650 (Bio-Techne, Wiesbaden, Germany) and the Opal fluorophore 780 (Akoya Biosciences, Marlborough, MA, USA) were used for signal detection. Nuclei were counterstained with 4′,6-diamidino-2-phenylindole (DAPI) provided with the Multiplex Fluorescent v2 kit. The slices were mounted with ProLong Gold Antifade Mountant (Thermo Fisher Scientific) and stored at 4°C until further analysis. RNAscope® probes used are listed in Table 3.

## Immunohistochemistry

To detect HIF-2 stabilization using immunohistochemistry, kidneys were perfusion-fixed with 3% paraformaldehyde and after dehydration in an ascending methanol and isopropanol series embedded in paraffin. HIF-2α (polyclonal rabbit anti-HIF-2α; dilution 1:5000; NB100-122; Bio-Techne) staining was performed on 5 μm sections. Sections were deparaffinized and pre-treated with Target Retrieval Solution (Agilent Technologies, Santa Clara, CA, USA) for 12 min in a pressure cooker. After cooling and washing in phosphate-buffered saline (PBS), the sections were blocked with avidin solution (Avidin/Biotin blocking kit, VectorLabs, Newark, CA, USA) twice for 20 min and then treated with 3% hydrogen peroxide for 10 min. After washing with PBS, the sections were blocked with Serum-free Protein Block (Agilent Technologies, Waldbronn, Germany) for 1 h. Sections were incubated at 4°C overnight with HIF-2α antibody diluted in 1% BSA/PBS. After three washes with PBS, sections were incubated with donkey-anti-rabbit-horseradish peroxidase (HRP)

**Table 3. RNAscope probes used for *in situ* hybridization**

| RNAscope® probe | Cat No. | RNAscope® probe | Cat no. |
|---|---|---|---|
| Mm-Adm | 493601 | Mm-Epo-C2 | 315501-C2 |
| Mm-Cebpd | 556661 | Mm-Pdgfrb | 411381 |
| Mm-Cdh16 | 582781 | Mm-Pdgfrb-C3 | 411381-C3 |
| Mm-Egln1 (PHD2) | 315491 | Mm-Rgs4-C3 | 467461-C3 |
| Mm-Egln3 (PHD3) | 434931 | Mm-Tcf21-C2 | 508661-C2 |
| Mm-Epas1-E7-E13 (HIF-2$\alpha$) | 450861 | Positive control probe | 321651 |
| Mm-Epo | 315501 | Negative control probe | 320751 |

(dilution 1:500; Cell Signaling Technology, Danvers, MA, USA) for 45 min at room temperature. After washing with PBS, sections were treated with the TSA Plus Biotin Kit (dilution 1:100, 15 min, Akoya Biosciences) for signal amplification, washed with PBS and incubated with streptavidin–HRP (Abcam, Cambridge, UK) for 30 min. Signals were detected with freshly prepared 3,3′-diaminobenzidine (DAB) solution (DAB Peroxidase Substrate Kit, VectorLabs). Sections were counterstained with haematoxylin, blued with ammonium–water and mounted with Dako Glycergel Mounting Medium (Agilent Technologies).

## Microscopy

All images were taken with an Axio Observer.Z1 Microscope (Zeiss, Jena, Germany) using the Plan-Apochromat $\times$20/0.8 objective and the Colibri7 as light source. Fluorescence images were captured with the Axiocam 506mono, brightfield images using the Axiocam 305. Filters used were the filter set 43-Cy3 (EX BP 545/25; EM BP 605/70), filter set 50-Cy5 shift free (EX BP 640/30; EM BP 690/50), filter set 96 HE BFP (EX BP 390/40; EM BP 450/40), and filter set 115-Cy7 (EX BP 710/87; EM BP 814/91) (Zeiss). For detailed fluorescence images, the Apotome.2 system (Zeiss) was used to take 10 to 12 z-stacked images, which were combined using maximum projection. Overviews were generated by combining tiles taken at $\times$20 magnification. Images in the same figure were taken with the same light intensities, exposure times and displayed with identical image modifications.

## Image analysis

The number of target mRNA$^+$ cells per kidney section or detail was determined using ImageJ software. Five kidneys were analysed per genotype. For each data point, target mRNA$^+$ cells were counted and labelled on three sections/details per kidney using the 'cell counter notice' tool and the mean values were calculated.

**Table 4. Scoring scheme**

| Score | |
|---|---|
| 0 | 0 dots/ PDGFR-$\beta^+$ cell |
| 1 | 1–5 dots/ PDGFR-$\beta^+$ cell |
| 2 | 6–10 dots/ PDGFR-$\beta^+$ cell |
| 3 | >10 dots/ PDGFR-$\beta^+$ cell without dot clusters |
| 4 | >10 dots/ PDGFR-$\beta^+$ cell with dot clusters |

In the RNAscope method, one signal point corresponds to one mRNA copy. Therefore, the expression level of a target mRNA per cell can be estimated from the number of RNAscope signals per cell, at least as long as the expression level per cell is not too high (results in clustered or extensive homogenous signals). To assess the target mRNA expression level per cell, if this method was applicable, five representative high magnification images (1000 px $\times$ 1000 px) per kidney zone were taken and the number of target mRNA signal dots per PDGFR-$\beta^+$ cell was manually scored according to the scheme shown in Table 4.

Kidneys of three different animals per condition/genotype were analysed. Counting/scoring was performed by two investigators independently, without knowledge of treatment or genotype.

Intellesis software (Zeiss) was used for automated image analysis to determine the coexpression of HIF-2$\alpha$ and PDGFR-$\beta$. First, the zonal boundaries were defined. Then HIF-2$\alpha$/PDGFR-$\beta$ co-expression was determined for each zone and normalized to PDGFR-$\beta$ expression/area. Segmentation was achieved through background subtraction with the rolling ball method to ensure the inclusion of all RNAscope signals. For PDGFR-$\beta$, pixel intensity values within the range of 1500 to the maximum measurable intensity (16,384) were included, with a tolerance level set at 3%. The rolling ball algorithm was employed for background subtraction, using a radius of 30 pixels to optimize segmentation accuracy. Similarly, for HIF-2$\alpha$, intensity values between 1500 and the maximum (16,384) were analysed under the same tolerance level of

3%, but with a rolling ball radius of 10 pixels to account for differences in signal characteristics. No size exclusion criteria were implemented during the analysis to ensure the inclusion of all detected RNAscope signals. A total of at least five representative images per genotype and experimental condition were processed and analysed.

To determine the percentage of PDGFR-$\beta^+$ fibroblasts that coexpress either PHD2 or PHD3 per kidney zone, Intellesis software (Zeiss) was used for automated image analysis. First, the zonal boundaries were defined. Then zones of interest (ZOIs) were defined using cell nuclei as reference points, with a defined radius around each nucleus to facilitate cell detection. mRNA transcripts were detected and quantified within these established ZOIs. For each signal, thresholds were set to detect all signals ranging from the minimum detection threshold to the maximum measurable intensity of 16,384, ensuring comprehensive signal capture across the entire dynamic range. No size exclusion criteria were implemented during the analysis to ensure the inclusion of all detected RNAscope signals. Then, the number of PHD2/PDGFR-$\beta^+$ or PHD3/PDGFR-$\beta^+$ ZOIs per zone was determined and set in relation to the total number of PDGFR-$\beta^+$ ZOIs in the same zone. A total of five representative images of normoxic wild-type kidneys were processed and analysed.

### Statistical analyses

All data are presented as means $\pm$ SD. Statistical significance was determined by one-way ANOVA with Tukey's correction or Dunnett's multiple comparisons as well as an unpaired Student's $t$ test, one-tailed as stated in the results section. $P$-values and group sizes are stated in the results section. $P \leq 0.05$ after correction for multiple testing was considered statistically significant. The data were analysed using GraphPad Prism10.4.0 (GraphPad Software, Boston, MA, USA).

### Results

#### HIF-2$\alpha$ can be stabilized in fibroblasts throughout all kidney zones by pharmacologic PHD inhibition

Under hypoxic conditions, such as anaemia or low ambient oxygen pressure, predominantly cortical fibroblasts are recruited for additional EPO production starting from the cortico-medullary border towards the cortex surface, although we could show that deeper medullary fibroblasts would in principle be able to produce EPO. Interestingly, when EPO expression is stimulated with a single dose of PHD inhibitors (PHDi), the recruitment pattern is very similar to that observed with physiological stimuli, although induction should

occur independently of tissue oxygenation. These contradictory findings raised the question of whether, in contrast to genetic HIF-2 activation, the physiological or pharmacological stimuli are not sufficient to stabilize HIF-2 in deeper medullary fibroblasts. Thus, we aimed to analyse the expression levels of HIF-2$\alpha$ mRNA, the stabilization of HIF-2$\alpha$ and the induction of HIF-2 target genes in cortical and medullary interstitial fibroblasts under different conditions.

HIF-2$\alpha$ mRNA expression in interstitial fibroblasts was investigated using the RNAscope multiplex assay. Kidney sections of wild-type mice were analysed under normoxic and anaemic conditions (haematocrit values (Hct) 25%). Additionally, to examine HIF-2$\alpha$ expression and stabilization independent of tissue oxygenation, kidneys from PHDi-treated wild-type mice were analysed. To ensure that a potential lack of HIF-2 stabilization could not be caused by insufficient dosage of PHDi, the mice received multiple doses of roxadustat at 90-min intervals. Maximal target induction was reached after eight applications (for detailed results see Fig. 4E). The kidneys were fixed and analysed 90 min after the last PHDi administration. If not stated otherwise kidneys of eight-times-treated mice were used for the respective analyses. Moreover, sections of PDGFR-$\beta^{\text{CreERT2/+}}$ Vhl$^{\text{ff}}$ mice were included in the study as a model for genetically stabilized HIF signalling.

In each of the conditions analysed, interstitial PDGFR-$\beta^+$ fibroblasts showed clear HIF-2$\alpha$ mRNA signals (Fig. 1A). Even higher HIF-2$\alpha$ mRNA expression levels per cell were detected in endothelial cells and intraglomerular mesangial cells. In addition, HIF-2$\alpha$ expression was detected in tubular cells, albeit at lower levels per cell. Manual scoring of HIF-2$\alpha$ signals per cell in PDGFR-$\beta^+$ interstitial fibroblasts revealed that, in each kidney zone, approximately 60–70% of the cells coexpressing HIF-2$\alpha$/PDGFR-$\beta$ could be assigned a score of 1 (1–5 HIF-2$\alpha$ signal dots per cell) under each condition; 25–30% of these cells had a score of 2 (6–10 signal dots), and about 5–10% had a score of 3 (>10 points without clusters). There were no differences in the HIF-2$\alpha$ expression levels in interstitial fibroblasts between kidney zones or conditions. Automated coexpression analysis using the Zeiss Intellesis software revealed also no significant differences in the interstitial coexpression of HIF-2$\alpha$ and PDGFR-$\beta$ between the different zones under the same condition or between one certain zone under different conditions. On average, HIF-2$\alpha$/PDGFR-$\beta$ co-expression accounted for 1.4–2.2% of the total area in the cortex across conditions. In the outer medulla, the percentage was 1.1–1.9%, and in the inner medulla, it was 0.8–1.5% (Fig. 1B). These results were consistent with those obtained from the manual analysis of the RNAscope images. Quantitative analysis of HIF-1$\alpha$ and HIF-2$\alpha$ mRNA abundances in the kidneys also revealed

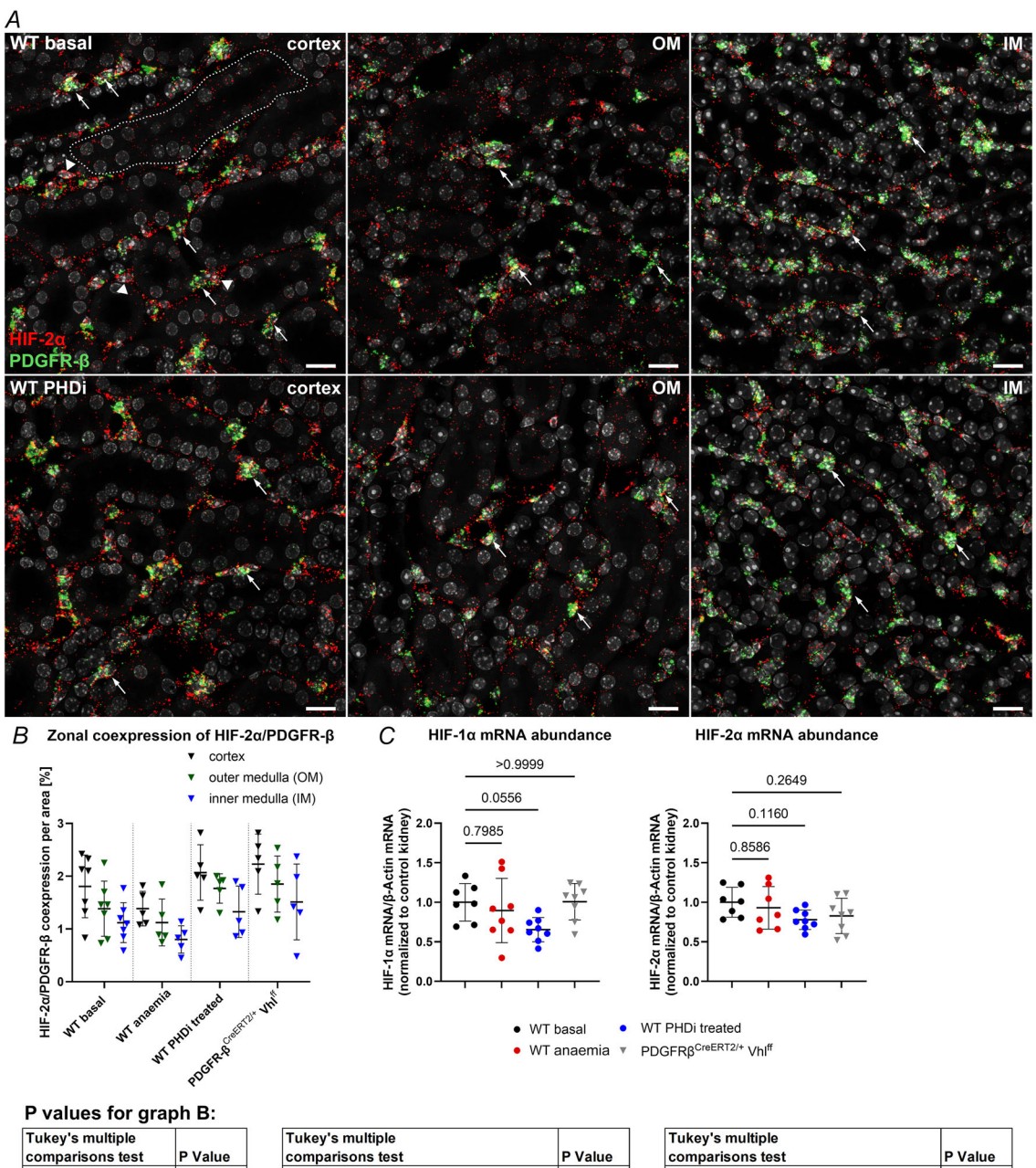

**Figure 1. Cellular HIF-2α mRNA expression levels as well as renal HIF-1α and HIF-2α mRNA expression under different conditions**

*A*, zonal details showing the co-expression of HIF-2α (red) and PDGFR-β (green) using RNAscope. The upper row shows details from the cortex, outer medulla (OM), and inner medulla (IM) of a kidney section of a wild-type (WT)

mouse under normoxic conditions. The lower row shows the respective details from a wild-type mouse treated 8 times with PHDi. Arrows highlight exemplarily some HIF-2$\alpha$/PDGFR-$\beta$ coexpressing fibroblasts. The arrowheads indicate some endothelial cells, and the dotted line indicates a tubular segment. Scoring the HIF-2$\alpha$ expression level per PDGFR-$\beta^+$ cell revealed no differences between kidney zones or different conditions. Clear HIF-2$\alpha$ signals were observed in endothelial cells, while only weak HIF-2$\alpha$ expression was detectable in tubular cells. Nuclei were counterstained with DAPI (grey). Scale bars: 20 µm. *B*, automated co-expression analysis revealed no significant differences in HIF-2$\alpha$/PDGFR-$\beta$ co-expression among kidney zones within a given condition, nor between any given zone and its corresponding zone in the other analysed conditions. Statistical significance was determined using one-way ANOVA with Tukey's multiple comparisons test. Respective *P*-values are provided in the tables below the graphs. Values are means ± SD of $n \geq 5$ per group. *C*, renal HIF-1$\alpha$ and HIF-2$\alpha$ mRNA abundances under normoxic, anaemic and PHDi (8×)-treated conditions and in PDGFR-$\beta^{\text{CreERT2/+}}$ Vhl$^{\text{ff}}$ mice. There was no significant difference between the analysed conditions. Statistical significance was determined using one-way ANOVA with Dunnett's multiple comparisons test. *P*-values are stated above the lines. Values are means ± SD of $n \geq 7$ per group.

no significant differences between the conditions or genotypes (Fig. 1*C*).

Stabilization of HIF-2$\alpha$ protein was then analysed by immunohistochemical staining for HIF-2 positive nuclei on kidney sections of mice under the different conditions. Under normoxic conditions HIF-2 stabilization was restricted to very few interstitial fibroblasts along the corticomedullary border (Fig. 2 upper row). In kidneys of anaemic mice an increase in interstitial HIF-2 positive cells could be detected in the cortex and in the outer zone of the outer medulla. However, there was no clear HIF-2 stabilization in the deeper medullary regions – the inner zone of the outer medulla and the inner medulla (Fig. 2 second row). For clarification, the outer zone of the outer medulla refers to the area containing the S3 segments of the proximal tubule. The inner zone/stripe of the outer medulla, on the other hand, refers to the part of the outer medulla that lies between the outer stripe and the inner medulla. In contrast, interstitial HIF-2 positive cells could be detected throughout all kidney zones in the PDGFR-$\beta$-cell-specific Vhl knockout mice (Fig. 2 bottom row). Interestingly, in kidneys of 8× PHDi-treated mice, HIF-2 stabilization in interstitial fibroblasts could be found not only in the cortex but also in all medullary regions. Overall, the HIF-2 stabilization pattern in the interstitial fibroblasts was very similar to that in the Vhl-KO mice. In addition, HIF-2 stabilization could also be detected in endothelial cells, juxtaglomerular cells of afferent arterioles, intraglomerular mesangial cells and vascular smooth muscle cells. It is noteworthy that HIF-2 stabilization in interstitial fibroblasts in the deeper medullary regions was already observed after three administrations of PHDi, albeit in a smaller number of cells (for details see Fig. 5*C*).

## HIF-2$\alpha$ is functionally active in medullary fibroblasts, but does not induce EPO expression

To confirm the functional activity of stabilized HIF-2, the spatial induction of different HIF-2 target genes was analysed under the different conditions. Adrenomedullin and regulator of G-protein signalling 4 (RGS4) have

previously been shown to be upregulated together with EPO under hypoxaemic conditions HIF-2-dependently (Broeker et al., 2020). Therefore, the transcription of these genes was used as a readout of activated HIF-2-dependent gene expression.

Under normoxic conditions, ADM was detected in few cortical interstitial fibroblasts. Low levels of ADM were also expressed by tubular cells (Fig. 3*A*; see also Fig. 5*A*). In the kidneys of anaemic mice, ADM was upregulated in interstitial fibroblasts along the corticomedullary border, in most cases together with EPO, but also in some EPO-negative interstitial fibroblasts. In addition, ADM was increased in some tubular cells. PHDi treatment (8×) resulted in a strong upregulation of ADM expression in most interstitial fibroblasts of all kidney zones. Moreover, PHDi also significantly increased ADM expression in tubular cells (Fig. 3*A*). Due to the genetic stabilization of HIF-2 in PDGFR-$\beta^{\text{CreERT2/+}}$ Vhl$^{\text{ff}}$ mice, ADM was also detected in most PDGFR-$\beta^+$ cells in all renal zones. The interstitial ADM expression patterns were very similar in PHDi-treated and Vhl-KO mice.

Analysis of the RGS4 expression pattern showed very similar results regarding its induction in interstitial cells in the studied conditions. Under normoxic conditions, RGS4 was mainly found in the few EPO$^+$ cells and in a fibroblast cluster around these EPO$^+$ cells along the corticomedullary border. In addition, PDGFR-$\beta^+$ cells along the vasa recta and vascular smooth muscle cells were positive for RGS4. Under anaemic conditions, RGS4 was upregulated together with EPO in a clustered fashion along the corticomedullary border and in the cortex. In the deeper zones of the medulla, the expression of RGS4 was unchanged. In both, the PHDi-treated and Vhl-KO mice, RGS4 was induced in most fibroblasts across all renal zones (Fig. 3*A*).

Quantitative real-time PCR analysis confirmed the strong upregulation of ADM and RGS4 in the kidneys of PHDi-treated mice as well as the Vhl-KO mice (Fig. 3*B*).

In contrast to the expression of ADM and RGS4, EPO expression in PHDi-treated mice remained restricted to the cortex and the corticomedullary border including the outer stripe of the outer medulla (Fig. 3*A*), despite

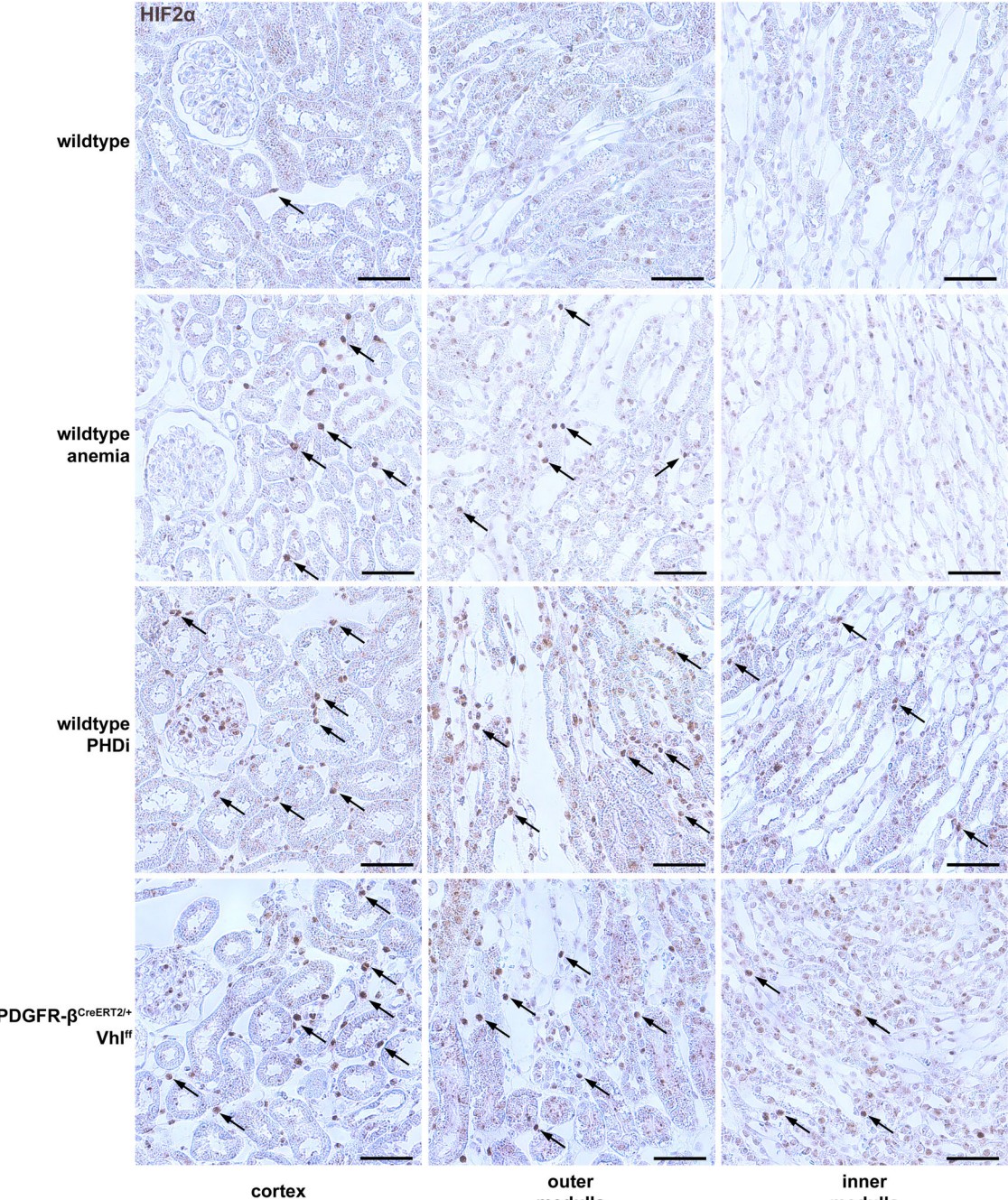

**Figure 2. HIF-2 protein stabilization on kidney sections under different conditions**

HIF-2 stabilization was visualized using immunohistochemical staining for HIF-2$\alpha$ (brown nuclear signal) on kidney sections of wild-type mice under different conditions as well as PDGFR-$\beta^{CreERT2/+}$ Vhl$^{ff}$ mice. For each condition/genotype, details from the cortex, outer medulla and inner medulla are shown. The arrows exemplarily highlight some HIF-2 positive interstitial fibroblasts. Under normoxic conditions only very few HIF-2 positive nuclei could be detected in the cortex (upper row). Under anaemic conditions clusters of HIF-2 positive fibroblasts could be detected in the cortex and in the outer zone of the outer medulla but not in the deeper medullary regions (second row). After 8× PHDi treatment most interstitial fibroblasts were positive for HIF-2 throughout all kidney zones (third row). On sections of PDGFR-$\beta^{CreERT2/+}$ Vhl$^{ff}$ mice HIF-2 positive interstitial fibroblasts could also be detected in all kidney zones (lower row). Scale bars: 50 μm.

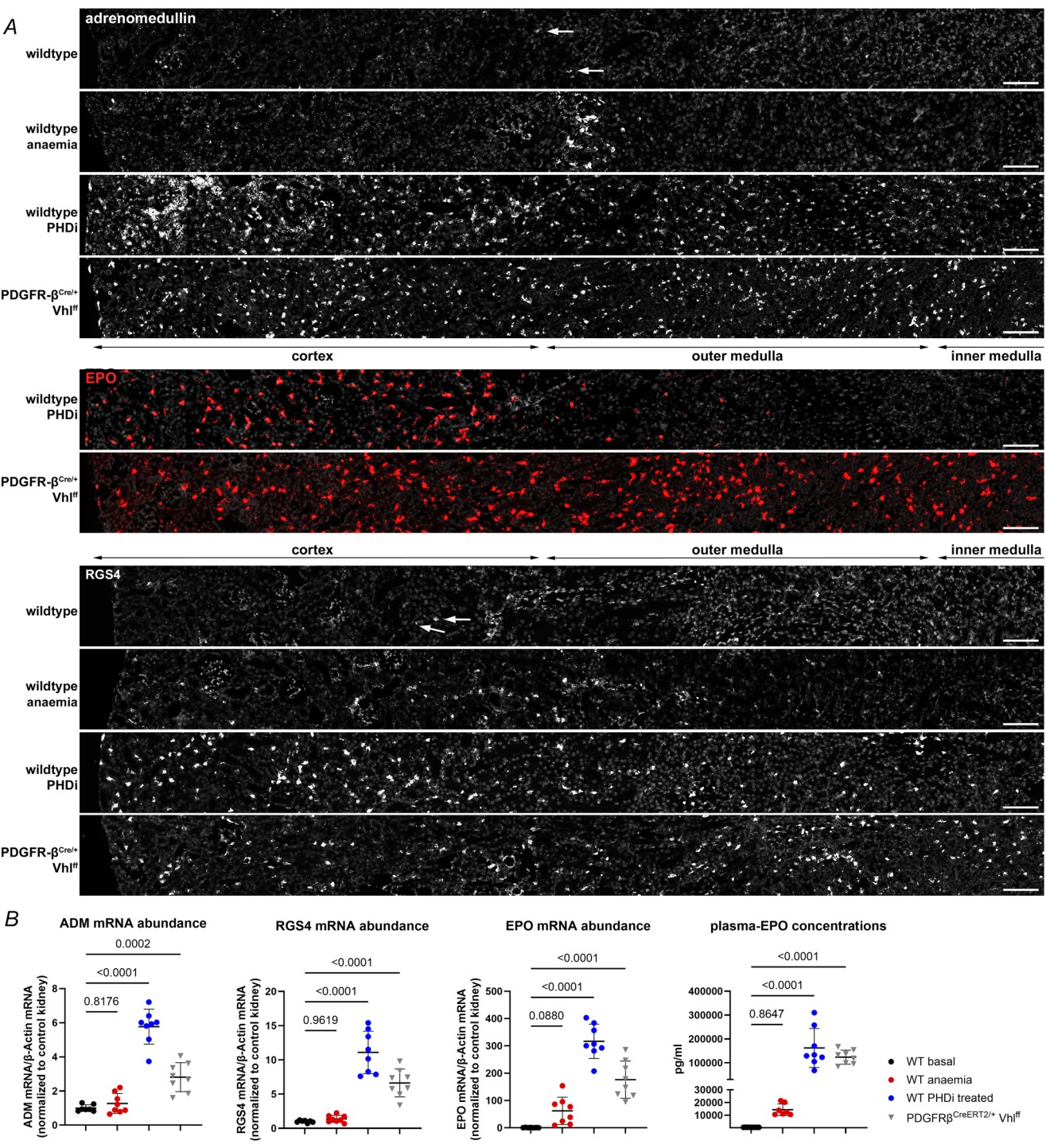

**Figure 3. Renal induction of different HIF-2 target genes under different conditions**
*A*, RNAscope for adrenomedullin (ADM, white) or RGS4 (white) on kidney sections of normoxic, anaemic, and 8× PHDi-treated wild-type mice compared to kidney sections of PDGFR-β^CreERT2/+ Vhl^ff mice. Under normoxic conditions ADM could only be detected sporadically in interstitial fibroblasts (arrows) and at a low level in some tubular segments. RGS4 could be detected in some interstitial fibroblasts along the corticomedullary border (arrows) and along vasa recta. ADM and RGS4 were upregulated in interstitial fibroblasts along the corticomedullary border in anaemic mice. After 8× PHDi treatment and in PDGFR-β^CreERT2/+ Vhl^ff mice, the expression patterns of ADM and RGS4 were quite similar. They could be detected in most fibroblasts across all kidney zones. In addition, ADM was upregulated in tubular segments of PHDi-treated mice. In contrast, EPO (red) could only be detected in

interstitial fibroblasts of the cortex and the outer zone of the outer medulla on kidney sections of PHDi-treated mice, while EPO expression was observed in most fibroblasts across all kidney zones in PDGFR-$\beta^{\text{CreERT2/+}}$ Vhl$^{\text{ff}}$ mice. Nuclei were counterstained with DAPI (grey). Scale bars: 500 μm. *B*, renal mRNA abundances of ADM, RGS4 as well as EPO and plasma EPO concentrations of mice under the analysed conditions/genotypes. Statistical significance was determined using one-way ANOVA with Dunnett's multiple comparisons test. *P* values are stated above the respective lines. Values are means ± SD of $n \geq 7$ per group.

HIF-2 stabilization in interstitial fibroblasts of the deeper medullary regions (see Fig. 2). EPO expression in the deeper medullary regions could only be detected after genetic HIF-2 stabilization through Vhl deletion (Fig. 3*A*; Table 5).

Additionally, we analysed the EPO expression patterns on wild-type kidneys resulting from an increasing number of PHDi-administrations. EPO induction started in fibroblast clusters along the corticomedullary border after a single dose of roxadustat (Fig. 4*A*). With three serial administrations of PHDi, EPO induction was predominantly increased in the outer zone of the outer medulla and in the deep and mid cortex (Fig. 4*B*). With additional PHDi applications the number of EPO producing cells further increased throughout the cortex including the outer cortex (Fig. 4*C* and *D*). As mentioned above, we could not detect EPO induction in the deeper medullary regions even after 8× PHDi. Interestingly, the induction pattern of EPO due to the increasing number of PHDi applications mirrored the induction pattern observed due to increasing hypoxaemic stimuli (Eckardt et al., 1993). qPCR analysis of EPO and the HIF-target genes ADM, RGS4 and PHD3 also showed an increasing induction of each of the target mRNAs with increasing numbers of PHDi administrations. Maximal induction was observed after eight roxadustat administrations (Fig. 4*E*). Further RNAscope analysis revealed that – in contrast to EPO – ADM induction could already be detected after three administrations of PHDi in fibroblasts from all renal zones, including the inner medulla/papilla (Fig. 5*B*; Table 5). Additional PHDi administrations further elevated the total number of ADM$^+$ fibroblasts throughout all kidney zones (see Fig. 3*A*).

### PHD3 deletion is not sufficient to induce renal EPO expression but augments the effect of PHD2 deletion

In the context of the metaplastic transformation of juxtaglomerular renin-producing cells into EPO-producing cells, we could show that PHD3 prevents induction of EPO in juxtaglomerular renin cells (Broeker et al., 2021). Thus, we wondered if PHD3 might also play a role in preventing EPO production in deeper medullary fibroblasts. To discern the functional roles of PHD2 and PHD3 for renal EPO expression, tamoxifen-inducible PDGFR-$\beta^{\text{CreERT2/+}}$ mice were used to cell-type-specifically delete PHD2, PHD3 or PHD2/PHD3 in interstitial fibroblasts. In each mouse line renal EPO mRNA abundances, kidney EPO

expression patterns as well as plasma EPO concentrations were determined. Since previous findings showed that Vhl deletion in PDGFR-$\beta^+$ cells induced EPO expression in most interstitial fibroblasts throughout all kidney zones, the strength of renal EPO induction in these PDGFR-$\beta^{\text{CreERT2/+}}$ Vhl$^{\text{ff}}$ animals served as reference.

Cell-type-specific deletion of PHD2 in interstitial fibroblasts resulted in a significant increase in EPO mRNA abundance to approximately 170-fold compared to control animals. In parallel, plasma EPO concentrations increased to a mean of 54,037.5 ± 23,406.1 pg/ml (controls: 212.4 ± 128.7 pg/ml) (Fig. 6), resulting in polycythaemia with haematocrit values of about 80% compared to about 50% in controls.

In contrast, deletion of PHD3 alone in PDGFR-$\beta^+$ cells had no significant effect on the renal EPO expression. Haematocrit values, plasma EPO concentrations and EPO mRNA abundance were statistically unchanged compared to controls (Fig. 6).

Codeletion of PHD2 and PHD3, however, significantly further increased the EPO production compared to PHD2 deletion alone. In PHD2/PHD3-deficient mice, renal EPO mRNA abundance was increased approximately 290-fold resulting in plasma EPO concentrations averaging 110399.8 ± 17159.9 pg/ml. These values were comparable to those obtained after Vhl deletion (340-fold EPO mRNA expression; plasma EPO concentrations: 103032.4 ± 15944.6 pg/ml) (Fig. 6).

### PHD2 deletion as well as PHD2/PHD3 deletion induces EPO expression throughout all kidney zones

To determine the number of recruited EPO-producing cells and their zonal distribution in the respective mouse models, a co-RNAscope for PDGFR-$\beta$ and EPO was analysed.

On transverse kidney sections from control mice and mice with PHD3 deletion in PDGFR-$\beta^+$ cells, 13.0 ± 5.5 EPO$^+$ and 16.6 ± 9.4 EPO$^+$ cells were detected along the cortico-medullary border, respectively (Fig. 7*A* and *B*). PHD2 deletion significantly increased the number of EPO-producing cells per transverse kidney section to a mean of 2620.4 ± 838.3 EPO$^+$ cells, while PHD2/PHD3 codeletion further elevated the number of EPO$^+$ cells to 5972.2 ± 337.9 per kidney section. Interestingly, the EPO cells were not restricted to the cortex and the cortico-medullary border in these genotypes. Even when only PHD2 was deleted, EPO$^+$ cells could be detected in

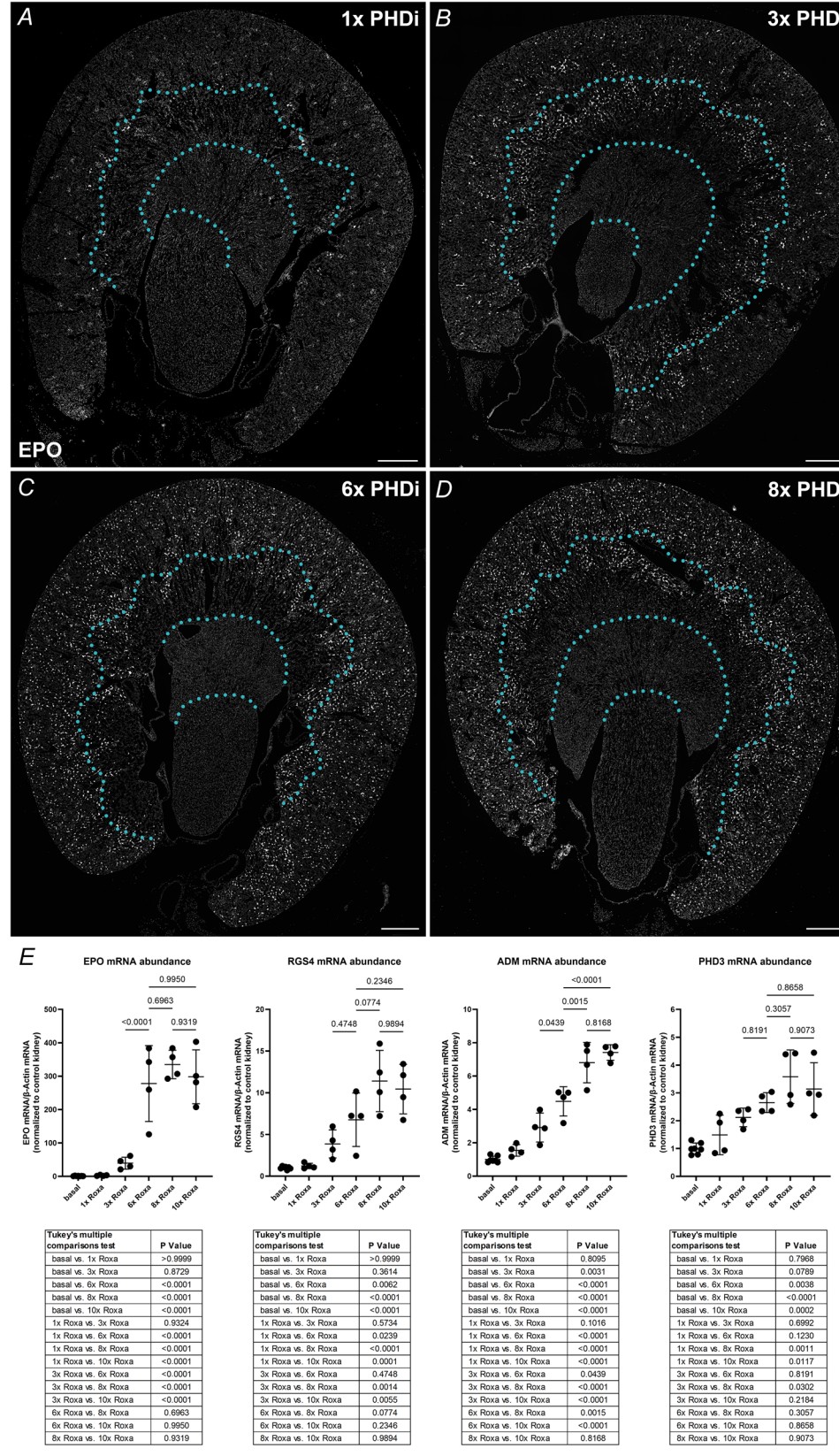

**Figure 4. Recruitment pattern of renal EPO mRNA expression as well as the renal expression levels of EPO mRNA and other HIF target genes after serial treatment of wild-type mice with the PHDi roxadustat**
*A–D,* distribution of EPO-producing cells on kidney sections of wild-type mice treated with 1× (*A*), 3× (*B*), 6× (*C*), or 8× (*D*) PHDi roxadustat was analysed using RNAscope for EPO mRNA (white). While 1× PHDi led to EPO

induction along the cortico-medullary border (*A*), EPO induction was increased in the outer stripe of the outer medulla and spread outward towards the cortex surface after 3× PHDi (*B*). With 6× and 8× PHDi the number of EPO producing cells further increased throughout the cortex. Dotted turquoise lines indicate the zonal borders. Nuclei were counterstained with DAPI (grey). Scale bars: 500 μm. *E*, renal mRNA abundances of different HIF-target genes – EPO, regulator of G protein signalling 4 (RGS4), adrenomedullin (ADM) and PHD3 – of wild-type mice treated with an increasing number of PHDi doses. Serial treatment with 10× roxadustat did not result in a further significant increase in target induction compared to an 8× series, as determined using one-way ANOVA with Tukey's multiple comparisons test. *P*-values are stated above the lines or in the tables below graphs. Values are means ± SD of $n \geq 4$ per genotype.

all kidney zones including the papilla. The additional deletion of PHD3 did not change the zonal distribution, but rather increased the density of EPO-producing cells per kidney zone (Fig. 7*C* and *D*).

Since interstitial PDGFR-$\beta^+$ cells themselves are not evenly distributed across all kidney zones (Gerl et al., 2016), the proportion of PDGFR-$\beta^+$ interstitial cells that coexpressed EPO was determined for each kidney

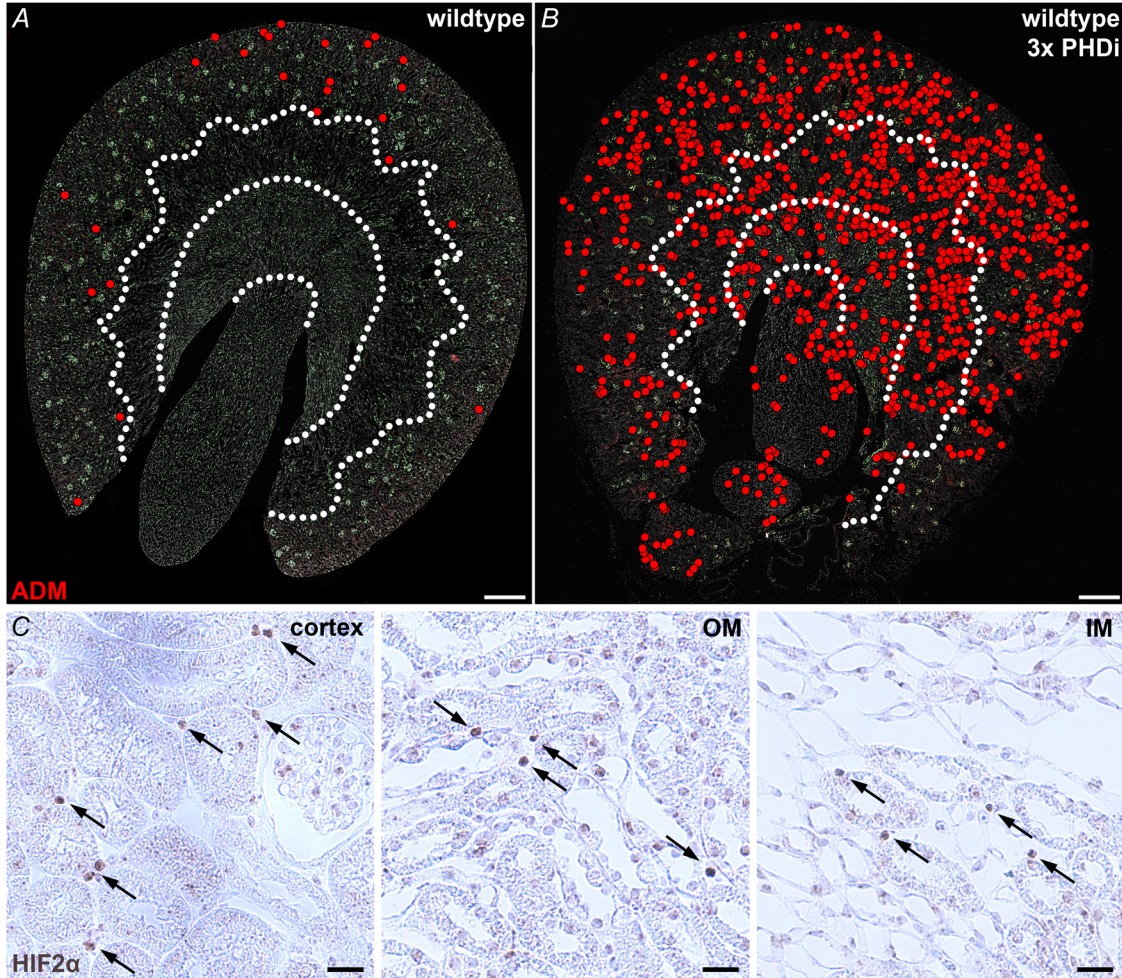

**Figure 5. Recruitment pattern of renal ADM expression as well as HIF-2 protein stabilization on kidney sections of wild-type mice 3× treated with the PHDi roxadustat**
*A* and *B*, distribution of ADM-producing cells (red dots) on kidney sections of normoxic wild-type mice and wild-type mice 3× treated with the PHDi roxadustat. Under normoxic conditions, ADM was only detected in few cortical interstitial fibroblasts (*A*). After 3 PHDi treatments, ADM$^+$ interstitial fibroblasts could be detected in a regular pattern throughout all kidney zones (*B*). Dotted white lines indicate the zonal borders. Nuclei were counterstained with DAPI (grey). Scale bars: 500 μm. *C*, HIF-2 stabilization was visualized using immunohistochemical staining for HIF-2$\alpha$ (brown nuclear signal) on kidney sections of wild-type mice after 3 PHDi treatments. The arrows highlight HIF-2 positive interstitial fibroblasts. After 3× PHDi treatment HIF-2 positive interstitial fibroblasts could be detected throughout all kidney zones – cortex, outer medulla (OM), and inner medulla (IM). Scale bars: 20 μm.

zone. In both PHD2- and PHD2/PHD3-deleted mice, most EPO-producing cells were found in the cortex and outer medulla. In PHD2-deficient mice, $47.2 \pm 7.5\%$ of PDGFR-$\beta^+$ interstitial cells were positive for EPO in the cortex and $49.3 \pm 5.0\%$ in the outer stripe of the outer medulla. In contrast, in mice with PHD2/PHD3 codeletion, $75.3 \pm 6.1\%$ of interstitial PDGFR-$\beta^+$ cells coexpressed EPO in the cortex and $75.9 \pm 5.4\%$ in the outer stripe of the outer medulla (Fig. 7*E* and *F*). The density of EPO expressing cells decreased significantly in both models in the inner stripe of the outer medulla, to $11.8 \pm 3.4\%$ and $36.7 \pm 3.7\%$, respectively. In the inner medulla, EPO$^+$ cells were only sporadically present. The overall distribution pattern of renal EPO producing cells in PDGFR-$\beta^{\text{CreERT2/+}}$ PHD2$^{\text{ff}}$ PHD3$^{\text{ff}}$ mice closely mirrored the distribution in PDGFR-$\beta^{\text{CreERT2/+}}$ Vhl$^{\text{ff}}$ mice (Broeker et al., 2020; Gerl et al., 2016).

## PHD3 impedes the induction of EPO in cortical fibroblasts

Based on the finding that deletion of PHD3 alone had no obvious effect on EPO expression, but significantly further increased the number of EPO-producing cells when PHD3 was codeleted with PHD2, we wondered whether PHD3 also attenuates/prevents EPO induction in a subset of fibroblasts in response to physiological or pharmacological stimuli. We therefore investigated whether HIF-2 stabilization by anaemia or PHDi treatment would alter the number of EPO$^+$ cells in PDGFR-$\beta^{\text{CreERT2/+}}$ PHD3$^{\text{ff}}$ mice compared to control animals. For this purpose, either anaemia with haematocrit values of about 25% was induced in PDGFR-$\beta$ cell-type specific PHD3-deficient mice and in control mice or the respective mice were treated with a series of three doses of the PHDi roxadustat.

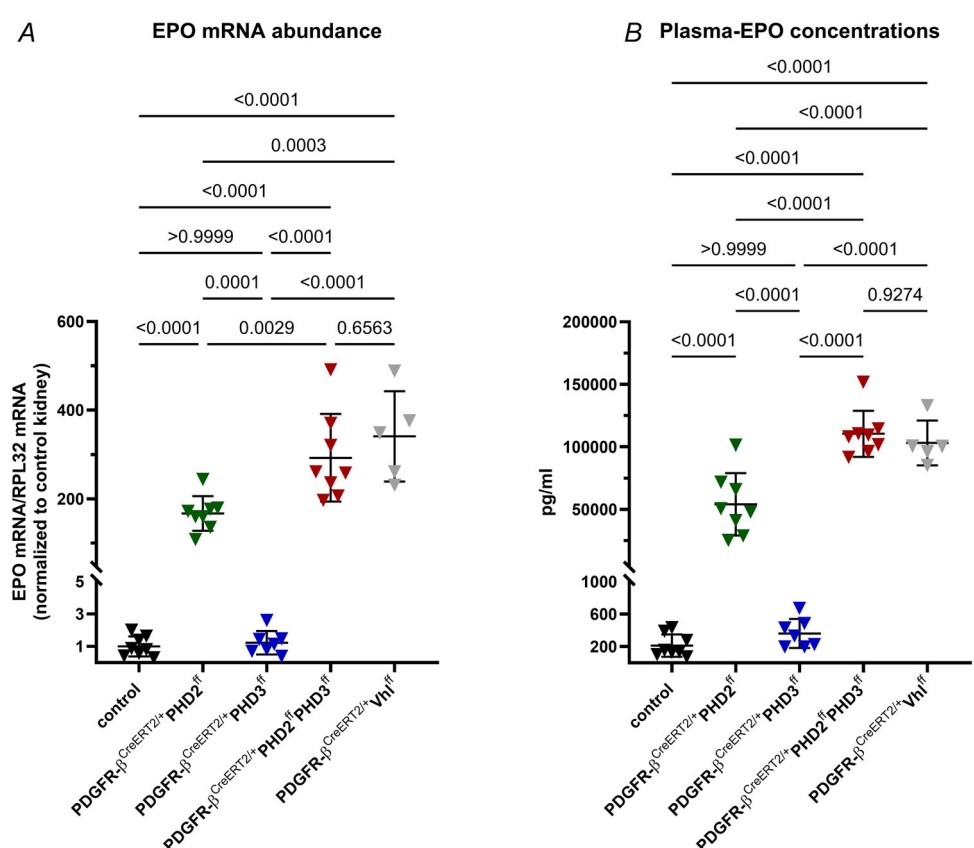

**Figure 6. Extent of renal EPO induction in mice with different PHD deletions or Vhl deletion in PDGFR-$\beta^+$ cells**
EPO mRNA abundances (*A*) and plasma EPO concentrations (*B*). While PHD3 deletion alone had no effect on the EPO expression, PHD2 deletion led to a significant increase in renal EPO mRNA expression levels and in parallel elevated plasma EPO concentrations of about 54,000 pg/ml compared to about 210 pg/ml in control animals. Combined deletion of PHD2 and PHD3 further increased the renal EPO mRNA abundance and plasma EPO concentrations to levels similar to mice with PDGFR-$\beta$-cell-specific Vhl deletion. Statistical significance was determined using one-way ANOVA with Tukey's correction. *P*-values are stated above the lines. Values are means $\pm$ SD of $n \geq 5$ per genotype.

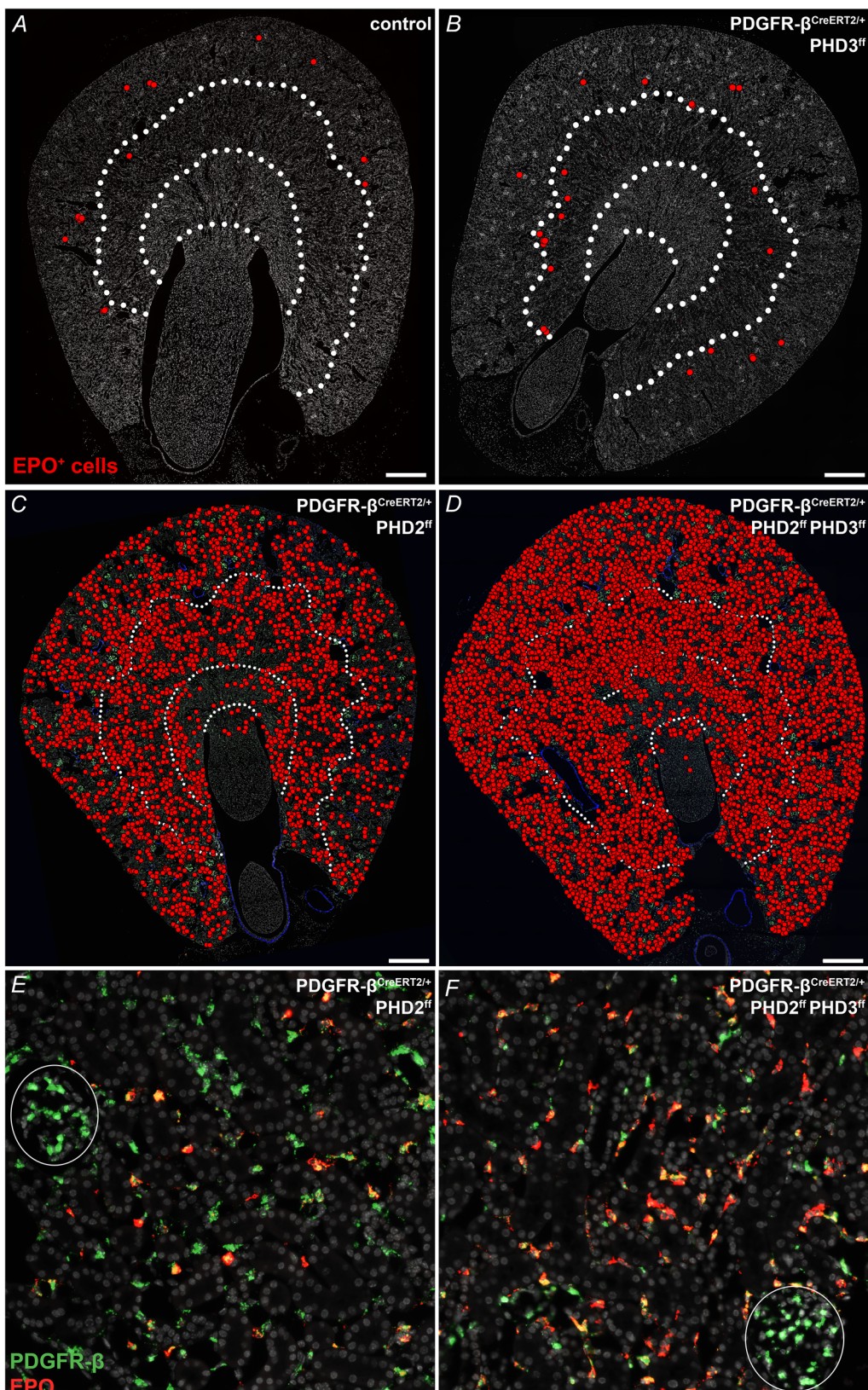

**Figure 7. Extent and distribution of renal EPO induction in mice with different PHD deletions in PDGFR-$\beta^+$ cells**

*A* and *B*, overviews of transverse kidney sections of control and PDGFR-$\beta^{CreERT2/+}$ PHD3$^{ff}$ mice showing the expression pattern of EPO producing cells (red dots). White dotted lines indicate zonal borders. Nuclei were

counterstained with DAPI (grey). Scale bars: 500 µm. *C* and *D*, overviews of transverse kidney sections showing the expression pattern of EPO producing cells (red dots) after PHD2 (*C*) or PHD2/PHD3 (*D*) deletion. Deletion of both, PHD2 or PHD2/PHD3 resulted in a strong induction of EPO throughout the kidneys. However, a direct comparison showed that the combined deletion of PHD2 and PHD3 resulted in approximately twice as many EPO$^+$ cells than PHD2 deletion alone. Dotted lines indicate zonal borders. Nuclei were counterstained with DAPI (grey). Scale bars: 500 µm. *E* and *F*, details from the kidney cortex of a co-RNAscope for EPO (red) and PDGFR-$\beta$ (green). After PHD2 deletion, EPO expression was induced in about 47% of PDGFR-$\beta^+$ interstitial fibroblasts (*E*). In contrast, codeletion of PHD2/PHD3 led to EPO production in about 75% of interstitial fibroblasts (*F*). Circles indicate glomeruli. Nuclei were counterstained with DAPI (grey). Scale bars: 50 µm.

Compared to the respective controls, both stimuli resulted in a significantly higher increase in renal EPO mRNA levels, plasma EPO concentrations as well as EPO$^+$ cells per kidney section in mice with PHD3 deletion. Specifically, anaemia in PHD3-KO animals resulted in twice the EPO mRNA abundance as in anaemic controls (28-fold *vs.* 13-fold increase). The same was observed for plasma EPO concentrations – anaemic PDGFR-$\beta^{CreERT2/+}$ PHD3$^{ff}$ mice had plasma EPO concentrations of 15783.2 $\pm$ 6015.3 pg/ml, while controls had a mean of 8725.9 $\pm$ 3181.6 pg/ml. In parallel, EPO cell numbers per kidney section were increased to about 600 $\pm$ 177 EPO$^+$ cells in PHD3-KO animals and 226 $\pm$ 131 EPO$^+$ cells in controls. In normoxic controls a mean of 13 $\pm$ 8 EPO$^+$ cells were detected (Fig. 8*A*–*C*).

Administration of three doses of PHDi even led to an 88-fold increase in renal EPO levels in PHD3-KO animals compared to normoxic controls, while a 23-fold increase was observed in PHDi-treated control animals. This resulted in plasma EPO concentrations of 33085.6 $\pm$ 13569.2 pg/ml in PHD3-KO mice and 8291.3 $\pm$ 2060.0 pg/ml in the respective controls. EPO cell numbers per transverse kidney section increased to 2559 $\pm$ 1035 and 897 $\pm$ 445 EPO$^+$ cells, respectively (Fig. 8*A*–*C*).

Of note, in control as well as PHD3-KO animals, recruitment of EPO-producing cells occurred only from the corticomedullary border towards the superficial cortex under both conditions (Fig. 8*D* and *E*).

Interestingly, when comparing renal EPO production under normoxia between PHD3-KO mice and their direct littermates from two litters, significantly higher EPO mRNA abundance and plasma EPO concentrations were detected in the PHD3-KO littermates. EPO expression levels were increased by approximately 40% and plasma EPO concentrations by approximately 80%. Overall, however, this did not lead to a significant increase in EPO cell numbers or in haematocrit values. EPO cell numbers were 14 $\pm$ 7 EPO$^+$ cells per section in controls and 24 $\pm$ 13 EPO$^+$ cells per section in PHD3-KO mice. Haematocrit values were 49.4 $\pm$ 1.9% in the controls and 47.9 $\pm$ 1.0% in the PHD3-KOs. The differences from the results shown in Fig. 6 are due to the different statistical test methods used and the fact that siblings from all mouse lines were used as control animals in Fig. 6.

## Expression pattern of PHD3 in fibroblasts differs from PHD2 but does not change under anaemic conditions or PHDi treatment

The PHD isoforms PHD2 and PHD3 have been reported to be themselves regulated in a HIF-dependent manner (D'Angelo et al., 2003; Marxsen et al., 2004; Stiehl et al., 2006). Thus, we wondered if hypoxaemic or pharmacological induced upregulation of PHD2 or PHD3 in deeper medullary fibroblasts could be a reason why EPO induction is restricted to the cortex. To investigate this hypothesis, the expression patterns of PHD2 and PHD3 were analysed in detail using RNAscope on kidney sections of wild-type mice under normoxic and anaemic conditions, as well as after pharmacological and genetic HIF stabilization. A particular focus was also placed on the extent of PHD3 and EPO coexpression.

First PHD2 and PHD3 expression was analysed on kidney sections of wild-type mice under normoxic conditions. PDGFR-$\beta$ was used as a marker for interstitial fibroblasts and cadherin 16 (Cdh16) was used as a marker for tubular cells. PHD2 positive RNAscope signals could be detected in all interstitial PDGFR-$\beta^+$ cells. Moreover, Cdh16$^+$ tubular cells were also positive for PHD2 (Fig. 9*A*). Automated RNAscope analysis using the Zeiss Intellesis software also showed that in each kidney zone about 90% of interstitial PDGFR-$\beta^+$ cells coexpressed PHD2 (Fig. 9*J*). Scoring the number of PHD2 signal dots per PDGFR-$\beta^+$ cell to assess cellular expression levels revealed that they were similar across the different kidney zones. In the cortex and outer medulla more than 90% of interstitial PHD2/PDGFR-$\beta$ coexpressing cells showed only 1–5 PHD2 signal dots per cell (score 1), while about 7.5% of the cells had a score of 2 (6–10 dots). In the inner medulla, over 98% of the PHD2/PDGFR-$\beta^+$ cells were assigned a score of 1 (Fig. 9*K*).

In contrast, PHD3 expression could only be detected in about 35% of interstitial PDGFR-$\beta^+$ in each kidney zone by automated PHD3/PDGFR-$\beta$ coexpression analysis (Fig. 9*J*). Additionally, PHD3 was detectable in tubular cells (Fig. 9*C*). As observed for PHD2, the PHD3 expression level per interstitial PDGFR-$\beta^+$ cell was quite similar across renal zones. In the cortex and outer medulla about 95% of interstitial PHD3/PDGFR-$\beta$ coexpressing

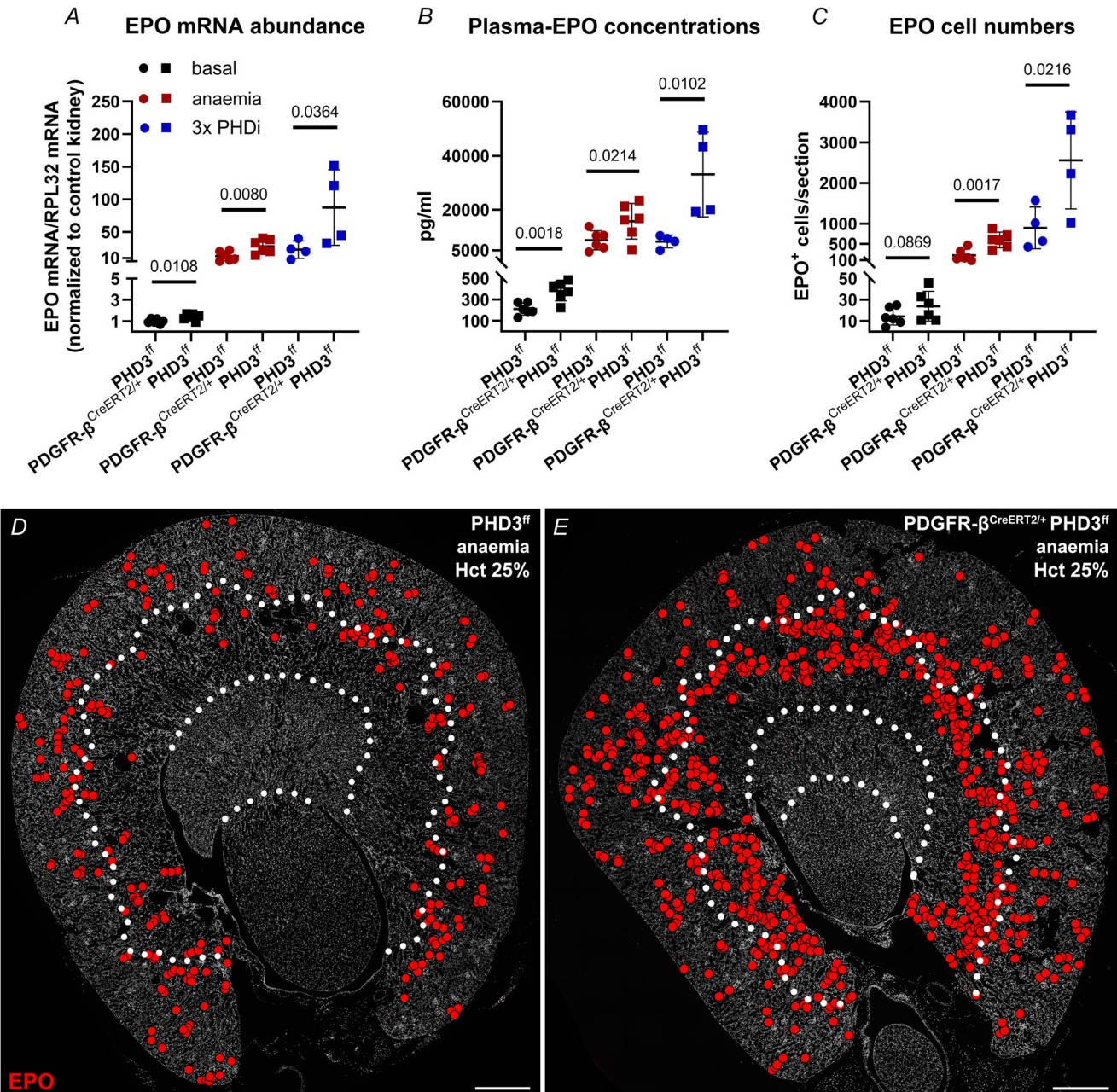

**Figure 8. Extent and distribution of renal EPO induction in PHD3-KO mice and controls under basal, anaemic and PHDi-treated (3×) conditions**

*A–C*, EPO mRNA abundance, plasma EPO concentrations as well as EPO cell numbers. The induction of EPO due to anaemia or 3× PHDi treatment was higher in PDGFR-$\beta$^CreERT2/+ PHD3^ff mice compared to controls. Statistical significance between control and PHD3-KO mice for each condition was determined using unpaired one-tailed *t* test. *P*-values are stated above the lines. Values are means ± SD of *n* ≥ 4 per genotype. *D* and *E*, overviews showing the expression pattern of EPO producing cells (red) on transverse kidney sections of control and PDGFR-$\beta$^CreERT2/+ PHD3^ff mice under anaemic conditions. Both animals had haematocrit values of approximately 25%. Recruitment of additional EPO cells followed the classic pattern from the corticomedullary border into the cortex. However, the total number of EPO^+ cells per kidney section was higher in the kidneys of PHD3-KO animals. White dotted lines indicate zonal borders. Nuclei were counterstained with DAPI (grey). Scale bars: 500 μm.

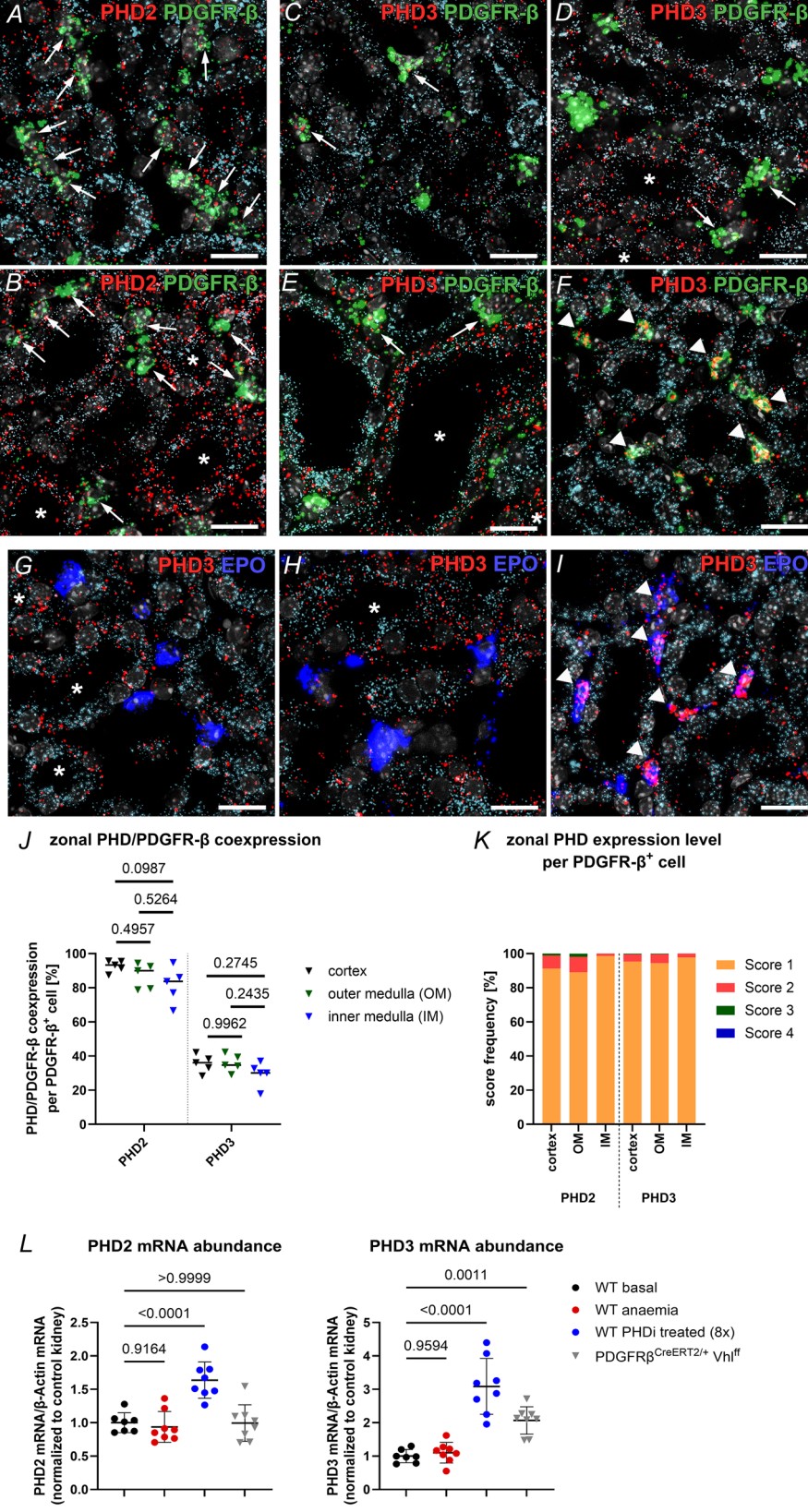

**Figure 9. Localization and regulation of PHD2 and PHD3 under basal, anaemic and PHDi-treated conditions as well as in PDGFR-$\beta^{\text{CreERT2/+}}$ Vhl$^{\text{ff}}$ mice**

*A* and *B*, details showing the colocalization of PHD2 (red) with PDGFR-$\beta$ (green) and the tubular marker cadherin 16 (light blue) under normoxic conditions (*A*) and after 8× PHDi treatment (*B*) using RNAscope. PHD2 could be

detected in all PDGFR-$\beta^+$ fibroblasts (arrows) as well as tubular cells under normoxia. After treatment with the PHD-inhibitor roxadustat, PHD2 expression was elevated in some tubular cells but not in interstitial fibroblasts (arrows). Asterisks highlight tubular cells with increased PHD2 expression compared to the controls. Nuclei were counterstained with DAPI (grey). Scale bars: 20 μm. C–F, RNAscope details showing the colocalization of PHD3 (red) with PDGFR-$\beta$ (green) and the tubular marker cadherin 16 (light blue) on kidney sections of wild-type mice under normoxic (C), anaemic (D) or 8× PHDi-treated (E) conditions and of PDGFR-$\beta^{CreERT2/+}$ Vhl$^{ff}$ (F) mice. Under normoxic conditions PHD3 was only expressed in about 35% of interstitial PDGFR-$\beta^+$ fibroblasts (arrows). In addition, PHD3 could be detected in some tubular segments. Under anaemic conditions and after PHDi treatment, PHD3 was only upregulated in some tubular segments, highlighted with asterisks. There was no upregulation of PHD3 in interstitial fibroblasts. Upregulation of PHD3 in interstitial fibroblasts could only be detected in Vhl-KO mice (arrowheads). Nuclei were counterstained with DAPI (grey). Scale bars: 20 μm. G–I, RNAscope details showing the colocalization of PHD3 (red) with EPO (blue) and the tubular marker cadherin 16 (light blue) on kidney sections of wild-type mice under anaemic (G) or 3× PHDi-treated (H) conditions and of PDGFR-$\beta^{CreERT2/+}$ Vhl$^{ff}$ (I) mice. In anaemic mice and mice treated with three administrations of PHDi, EPO was almost exclusively detected in PHD3 negative fibroblasts. In Vhl-KO mice a strong coexpression of PHD3 and EPO could be observed (arrowheads). Nuclei were counterstained with DAPI (grey). Scale bars: 20 μm. J, zonal PHD2/PDGFR-$\beta$ as well as PHD3/PDGFR-$\beta$ coexpression was determined on kidney sections of wild-type mice under normoxic conditions with the Zeiss Intellesis software. Analysis showed that in each kidney zone about 90% of interstitial fibroblasts were positive for PHD2, while only about 35% of the interstitial PDGFR-$\beta^+$ fibroblasts coexpressed PHD3. Statistical significance was determined using one-way ANOVA with Tukey's multiple comparisons test. P-values are stated above the respective lines. Values are means ± SD of n = 5 animals. K, zonal PHD2 or PHD3 mRNA expression level per PDGFR-$\beta^+$ fibroblast was determined on kidney sections of wild-type mice under normoxic conditions by manual scoring. All PHD2/PDGFR-$\beta^+$ or PHD3/PDGFR-$\beta^+$ fibroblasts per renal zone respectively constitute 100%. Score 1: 1–5 PHD2 or PHD3 signal dots per PDGFR-$\beta^+$ fibroblast; score 2: 5–10 signal dots; score 3: >10 dots without dot clusters. The PHD2 or PHD3 expression levels per PDGFR-$\beta^+$ fibroblast were quite similar between the different kidney zones. Values are means ± SD of n = 5 animals. L, renal mRNA abundances of PHD2 and PHD3 under different conditions. Statistical significance was determined using one-way ANOVA with Dunnett's multiple comparisons test. P-values are stated above the respective lines. Values are means ± SD of n ≥ 7 per group.

cells were assigned a score of 1, while about 4–5% of the cells had a score of 2. In the inner medulla, about 98% of the PHD3/PDGFR-$\beta^+$ cells showed a score of 1 (Fig. 9K).

Regarding the HIF-dependent regulation of PHD isoforms, qPCR revealed a significant upregulation of renal PHD2 in PHDi-treated (8×) mice, as well as a significant upregulation of PHD3 after PHDi treatment (8×) and after Vhl deletion (Fig. 9L). PHD2 upregulation due to PHDi treatment could be observed in tubular cells, while the PHD2 expression level in interstitial PDGFR-$\beta^+$ fibroblasts remained unchanged (Fig. 9B). In parallel, also increased PHD3 expression per cell could be detected in some tubular segments after PHDi treatment (8×). PHD3 upregulation was detectable neither in cortical nor deeper medullary fibroblasts (Fig. 9E). In contrast, in Vhl-KO mice PHD3 upregulation could exclusively be detected in interstitial fibroblasts (Fig. 9F). Under anaemic conditions a slight upregulation of PHD3 could be observed in some tubular segments, but not in interstitial fibroblasts (Fig. 9D).

Coexpression analysis of EPO and PHD3 showed that EPO induction almost exclusively occurred in PHD3 negative interstitial fibroblasts under anaemic conditions (Fig. 9G). Similarly, after three doses of PHDi, EPO induction was almost exclusively detected in PHD3 negative fibroblasts (Fig. 9H). Only after six or eight doses of PHDi, EPO was also induced in PHD3$^+$ interstitial fibroblasts of the cortex and the outer zone of the outer medulla. In Vhl-KO mice a strong coexpression of PHD3 and EPO could be observed in most EPO-producing cells (Fig. 9I).

## The expression patterns of the potential EPO regulating cofactors Tcf21, Cebpd and Kdm3a do not differ between cortical and medullary fibroblasts

Based on the finding that EPO was not expressed in deeper medullary fibroblasts after PHDi treatment, although HIF-2 signalling was active, and the finding that the PHD2 and PHD3 expression pattern could not account for the differential expression of EPO in cortical and deeper medullary fibroblasts, we considered other factors that might influence HIF signalling or EPO gene transcription.

For example, cortical and medullary fibroblasts might differ in the expression of co-transcription factors that are important for EPO expression. In addition, chromatin accessibility, i.e. different methylation and acetylation patterns, could be different between cortical and medullary fibroblasts.

Recently, two transcriptional cofactors have been reported as potential EPO regulators – Tcf21 (transcription factor 21) and Cebpd (CCAAT/enhancer-binding protein delta) (Kragesteen et al., 2023; Mao et al., 2023). Therefore, we analysed the zonal expression pattern of these factors in our respective mouse models.

RNAscope analysis showed that Tcf21 was expressed almost exclusively in interstitial PDGFR-$\beta^+$ cells in

**Table 5. Overview of the colocalization of the analysed targets in interstitial PDGFR-$\beta^{+}$ fibroblasts across all renal zones under different conditions**

| | Wild-type | Wild-type anaemia | Wild-type 3× PHDi | Wild-type 8× PHDi | PDGFR-$\beta^{Cre}$ Vhl$^{ff}$ |
|---|---|---|---|---|---|
| Cortex | HIF-2 protein, EPO, RGS4, ADM, Cebpd<br>Tcf21<br>PHD2<br>PHD2 PHD3 (35%) | HIF-2 protein, EPO, RGS4, ADM, Cebpd<br>Tcf21 | HIF-2 protein, EPO, RGS4, ADM, Cebpd<br>Tcf21 | HIF-2 protein, EPO, RGS4, ADM, Cebpd | HIF-2 protein, EPO, RGS4, ADM, Cebpd |
| Outer stripe of outer medulla | Tcf21<br>PHD2<br>PHD2 PHD3 (35%) | HIF-2 protein, EPO, RGS4, ADM, Cebpd<br>Tcf21 | HIF-2 protein, EPO, RGS4, ADM, Cebpd<br>Tcf21 | HIF-2 protein, EPO, RGS4, ADM, Cebpd | HIF-2 protein, EPO, RGS4, ADM, Cebpd |
| Inner stripe of outer medulla | Tcf21<br>PHD2<br>PHD2 PHD3 (35%) | Tcf21, Cebpd | HIF-2 protein, RGS4, ADM, Cebpd<br>Tcf21 | HIF-2 protein, RGS4, ADM, Cebpd | HIF-2 protein, EPO, RGS4, ADM, Cebpd |
| Inner medulla | Tcf21<br>PHD2<br>PHD2 PHD3 (35%) | Tcf21, Cebpd | HIF-2 protein, RGS4, ADM, Cebpd<br>Tcf21 | HIF-2 protein, RGS4, ADM, Cebpd | HIF-2 protein, EPO, RGS4, ADM, Cebpd |

Each zone is subdivided into two types of interstitial fibroblasts: PHD2+ and PHD2/PHD3+ fibroblasts. The latter account for approximately 35% of the fibroblasts in each zone and are evenly distributed. ADM, adrenomedullin; Cebpd, CCAAT/enhancer-binding protein delta; EPO, erythropoietin; HIF-2, hypoxia-inducible factor 2; PHD, prolyl-4-hydroxylase; PHDi, prolyl-4-hydroxylase inhibitor; RGS4, regulator of G-protein signalling 4; Tcf21, transcription factor 21; Vhl, von-Hippel–Lindau protein.

the normoxic kidney (Fig. 10*A*). Podocytes were also positive for Tcf21, while PDGFR-$\beta^{+}$ intraglomerular mesangial cells did not express Tcf21. There was no difference in the level of Tcf21 expression per interstitial fibroblast between the renal zones under normoxia. On kidney sections of anaemic mice, Tcf21 expression was downregulated in fibroblasts that expressed EPO as well as in some EPO-negative interstitial fibroblasts in the cortex and along the corticomedullary border. In PDGFR-$\beta^{+}$ cells of the outer and inner medulla Tcf21 expression was unchanged compared to normoxic conditions. After PHDi treatment Tcf21 was strongly downregulated in EPO$^{+}$ cortical fibroblasts. Moreover, it was also clearly downregulated in deeper medullary fibroblasts negative for EPO. Similar results were obtained for PDGFR-$\beta^{CreERT2/+}$ Vhl$^{ff}$ animals; here Tcf21 was also downregulated in EPO$^{+}$ fibroblasts throughout all kidney zones (Fig. 10*A*; Table 5).

Cebpd, on the other hand, was hardly detectable in interstitial fibroblasts under normoxic conditions. Only a few Cebpd$^{+}$ PDGFR-$\beta^{+}$ cells were found in the outer zone of the outer medulla. Clear Cebpd expression could only be detected in tubuli located in the outer zone of the outer medulla (Fig. 10*B*). In kidneys of anaemic mice, a slight Cebpd expression could be detected in some, but not all interstitial fibroblasts. EPO$^{+}$ fibroblasts were always positive for Cebpd, but not all Cebpd$^{+}$ fibroblasts expressed EPO. Furthermore, the expression of Cebpd in some interstitial cells was not restricted to the cortex but could also be detected in the entire outer medulla and partly in the inner medulla. PHDi administration led to a clear upregulation of Cebpd expression, not only in all PDGFR-$\beta^{+}$ fibroblasts and mesangial cells, but also in the tubular system (S3 segment). In the Vhl-KO kidneys, Cebpd was also detected in all interstitial fibroblasts and mesangial cells (Fig. 10*B*; Table 5). Also for the histone demethylase Kdm3a (lysine demethylase 3A), which is reported as a co-activator of EPO expression under hypoxia (Tian et al., 2019), we could not detect any differences in the expression pattern between cortical and medullary fibroblasts.

To gain further insights into the differences between cortical and deeper medullary fibroblasts, and in particular to detect possible differences in epigenetic regulation that could explain the different findings regarding EPO induction under physiological/pharmacological and genetic HIF-2 stabilization, cortical and medullary fibroblasts will be isolated in further experiments and analysed in detail using, e.g. RNAseq and ATACseq techniques.

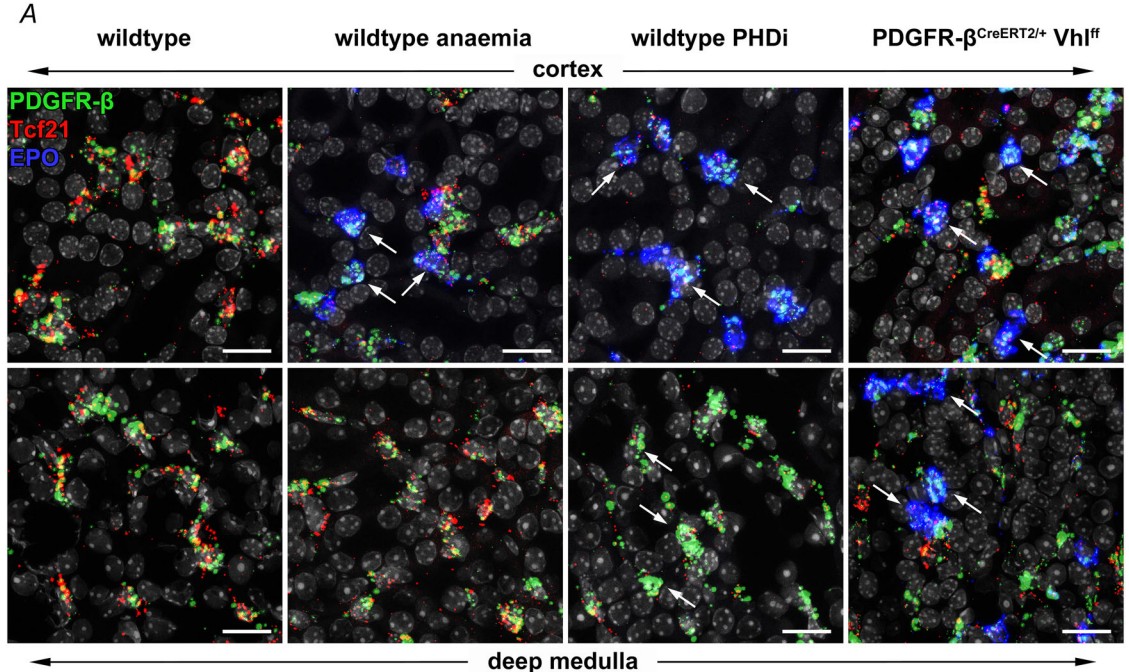

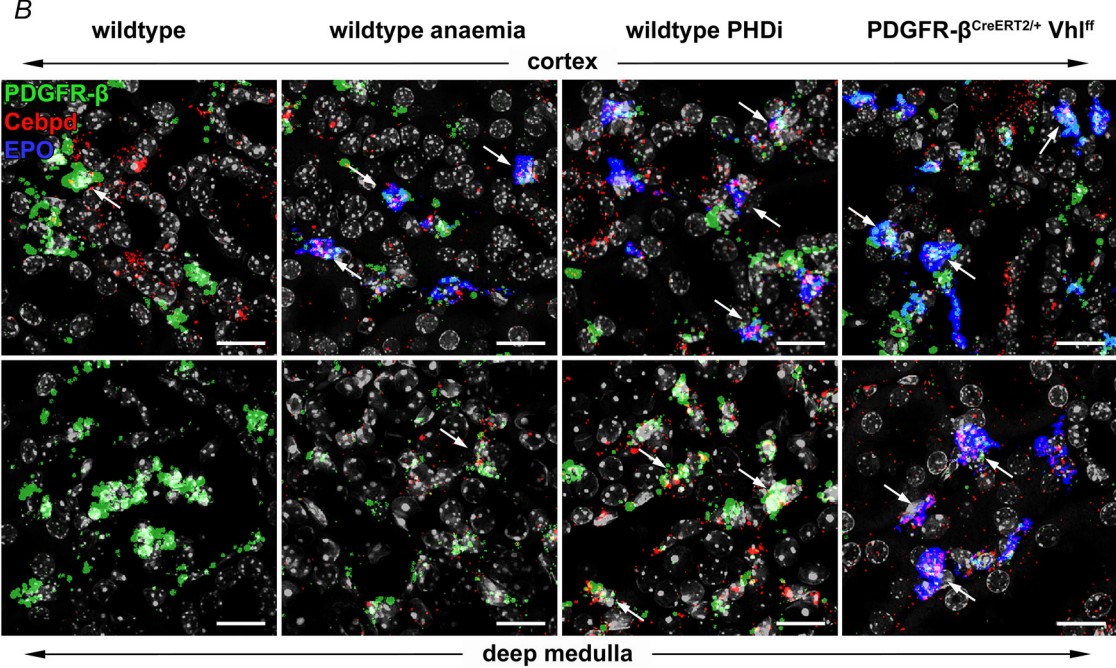

**Figure 10. Renal mRNA coexpression of the transcription factors Tcf21 (red) or Cebpd (CCAAT/ enhancer-binding protein delta, red), EPO (blue) and PDGFR-$\beta$ (green) under different conditions**
For each analysed condition a representative detail from the cortex and from the deeper medulla are shown. Nuclei were counterstained with DAPI (grey). Scale bars: 20 µm. *A*, under normoxic conditions Tcf21 could be detected in all interstitial fibroblasts. There was no difference in the expression level per cell between the different zones. Under anaemic conditions Tcf21 was significantly downregulated in EPO+ fibroblasts. Tcf21 expression in the deep medulla was unchanged compared to normoxic conditions. Under PHDi-treated conditions, Tcf21 expression was strongly downregulated in interstitial (EPO+) fibroblasts of the cortex and outer zone of the outer medulla. In the deeper medulla Tcf21 expression in EPO negative fibroblasts was also downregulated to levels comparable

to cortical fibroblasts. In PDGFR-$\beta^{CreERT2/+}$ Vhl$^{ff}$ animals Tcf21 was downregulated in all EPO$^+$ fibroblasts across all kidney zones. Arrows highlight some of the interstitial (EPO$^+$) fibroblasts, in which Tcf21 is downregulated. B, under normoxic conditions Cebpd was only sporadically detected in some interstitial fibroblasts along the cortico-medullary border. Moreover, Cebpd expression could be detected in the S3 segment of proximal tubules. Under anaemic conditions Cebpd was detectable in EPO$^+$ fibroblasts and some additional interstitial fibroblasts in the cortex and even in the deeper medulla. Under PHDi-treated conditions, clear Cebpd expression was found in most interstitial (EPO$^+$) fibroblasts of the cortex and outer zone of the outer medulla. In the deeper medulla Cebpd expression in EPO negative fibroblasts was also upregulated to levels comparable to cortical fibroblasts. PDGFR-$\beta^{CreERT2/+}$ Vhl$^{ff}$ animals showed similar Cebpd expression in interstitial fibroblasts across all kidney zones as PHDi-treated mice. Arrows highlight some of the interstitial Cebpd$^+$ (EPO$^+$) fibroblasts.

## Discussion

This study focused on the question of why deeper medullary fibroblasts are not involved in EPO production, although they are in principle capable of doing so. To this aim different hypotheses have been investigated. Firstly, the expression, stabilization and activity of the EPO-regulating transcription factor HIF-2$\alpha$ was investigated in order to examine whether the physiological hypoxaemic or pharmacological stimuli, in contrast to genetic interventions, may not be sufficient to stabilize HIF-2 in the deeper medullary fibroblasts. Secondly, the functional role of the HIF-2$\alpha$ regulating prolyl-4-hydroxylases PHD2 and PHD3 in the regulation of EPO was analysed, as well as the differential expression profiles of the two PHD isoforms in cortical and medullary fibroblasts. In addition, a potential upregulation of PHD2 and PHD3 due to the activation of HIF signalling was investigated.

One important finding of our study was that HIF-2$\alpha$ could indeed be stabilized in interstitial fibroblasts of the deeper medullary regions, including the papilla, without genetic intervention. However, HIF-2$\alpha$ stabilization was not sufficient to induce EPO in these cells. While anaemia with haematocrit levels of about 25% was apparently not a sufficiently strong hypoxic stimulus to stabilize HIF-2$\alpha$ in deeper medullary fibroblasts, a clear HIF-2$\alpha$ stabilization in interstitial fibroblasts of all renal zones, including the papilla, could be detected after serial administration of PHDi (Fig. 2). The number and distribution of HIF-2$\alpha$ positive interstitial fibroblasts in these kidneys was very similar to the kidneys of PDGFR-$\beta^{CreERT2/+}$ Vhl$^{ff}$ mice, in which HIF-2$\alpha$ was genetically stabilized (Fig. 2). In addition, after PHDi treatment endothelial cells as well as some juxtaglomerular and vascular smooth muscle cells were positive for HIF-2$\alpha$, which fits with previous reports of HIF-2 expression in these cell types (Broeker et al., 2021; Jaśkiewicz et al., 2022; Rosenberger et al., 2002). Of note, despite the distinct stabilization of HIF-2$\alpha$ in deeper medullary fibroblasts, there was no EPO expression detectable in these cells using the highly sensitive RNAscope assay although other HIF-2 target genes like ADM and RGS4 were strongly induced (Fig. 3), confirming active HIF-2 signalling in these cells. While the induction of HIF-2-dependent genes, including

EPO, is usually very rapid (Dahl et al., 2022; Stiehl et al., 2006), medullary fibroblasts appear to be protected from induction of EPO at least up to 12 h after the first PHDi administration. In contrast, HIF-2 stabilization and induction of the HIF-2 target gene, ADM, could already be detected throughout all kidney zones after three doses of PHDi, i.e. 4.5 h after the first PHDi application (Fig. 5).

Consistent with the findings on HIF-2 stabilization and activity in cortical and medullary fibroblasts, we also did not detect any differences for HIF-2$\alpha$ at the transcriptional level. Analysis of the HIF-2$\alpha$ expression levels per cell as well as the automated determination of zonal HIF-2$\alpha$–PDGFR-$\beta$ coexpression revealed no significant differences either between the renal zones – cortex, outer medulla and inner medulla – or between the conditions analysed for this study – normoxia, anaemia, PHDi administration and Vhl deletion in PDGFR-$\beta^+$ cells. Accordingly, total renal HIF-2$\alpha$ mRNA expression was also unchanged between the different conditions (Fig. 1). These results are in line with a recently published report in which the expression of HIF-2$\alpha$ mRNA was not upregulated in vitro after exposure to hypoxia for up to 24 h (Jaśkiewicz et al., 2022).

Having ruled out a lack of HIF-2 stabilization, we wondered if a distinct expression pattern of PHD2 and PHD3, which are thought to be the essential PHD isoforms for regulating renal EPO expression (Gardie et al., 2014; Kobayashi et al., 2016; Minamishima et al., 2008), might explain the observed differences between cortical and medullary EPO induction. Indeed, deletion of PHD2 specifically in PDGFR-$\beta^+$ cells profoundly increased EPO production, which was approximately doubled by co-deletion of PHD3 with PHD2 (Fig. 6), indicating at least two subsets of PDGFR-$\beta^+$ fibroblasts – solely PHD2-positive cells and PHD2/PHD3-double positive cells. This finding is in accordance with the previous observation that blood loss further increased the number of EPO expressing cells already stimulated by PHD2 deletion from the FoxD1 compartment, thereby hinting at differentially regulated subpopulations of potential EPO expressing cells (Kobayashi et al., 2016). Moreover, it suggests an inhibitory effect of PHD3 on EPO expression, which we have previously also observed in the context of the metaplastic transformation of renin-producing cells into EPO-producing cells (Broeker

et al., 2021). In line with this, a direct comparison of PHD3-KO mice with their respective controls confirmed an attenuating effect of PHD3 on EPO expression under basal conditions (Fig. 8).

Interestingly, however, the induction of EPO after PHD2 deletion did not occur only in the cortex and along the cortico-medullary border, as is the case with hypoxaemic recruitment of EPO-producing cells (Broeker et al., 2020; Eckardt et al., 1993), but spread throughout all renal zones up to the inner medulla (Fig. 7C). The additional deletion of PHD3 did not change the general distribution pattern of EPO-expressing cells observed due to PHD2 deletion, but rather led to a significant increase in EPO cell density in all kidney zones (Fig. 7C–F). The extent of EPO induction and the expression pattern of EPO$^+$ cells after PHD2/PHD3 deletion mirrored the effect of Vhl deletion in PDGFR-$\beta^+$ cells (Gerl et al., 2016). This finding provides further evidence that PDGFR-$\beta^+$ fibroblasts from all kidney zones are in principle potential EPO producers. It also supports the hypothesis that long-term HIF-2 stabilization, rather than side effects of the Vhl deletion, is the cause of widespread EPO induction. Involvement of HIF-1$\alpha$ is unlikely because cell-type-specific Vhl/HIF-1$\alpha$ codeletion mirrored the phenotype of Vhl deletion with respect to EPO induction, whereas Vhl/HIF-2$\alpha$ codeletion completely abolished EPO induction (Gerl et al., 2016).

The existence of PHD2$^+$ as well as PHD2/PHD3$^+$ subtypes of PDGFR-$\beta^+$ interstitial fibroblasts was further confirmed by the expression patterns of the PHD isoforms detected by RNAscope. While all interstitial fibroblasts were positive for PHD2, our analysis showed that only about 35% of fibroblasts also expressed PHD3 under normoxic conditions with PHD2/PHD3$^+$ interstitial fibroblasts being evenly distributed across kidney zones (Fig. 9). This result is consistent with the induction pattern of EPO achieved by the PDGFR-$\beta$-specific PHD2 or PHD2/3 deletion. Furthermore, our RNAscope analysis showed that the interstitial expression levels of PHD2 or PHD3 did not differ across different kidney zones (Fig. 9K).

However, the existence and distribution of PHD2$^+$ and PHD2/PHD3$^+$ subpopulations did not provide evidence as to why deeper medullary fibroblasts are not involved in EPO production under hypoxaemic or pharmacological conditions. In this context, we could also not detect compensatory upregulation of PHD2 or PHD3 after HIF-2 stabilization under anaemia or PHD inhibition, neither in medullary fibroblasts as a possible counter-regulation to prevent EPO induction, nor in EPO$^+$ and EPO$^-$ fibroblasts in the cortex (Fig. 9). In contrast, we observed a HIF-dependent increase in PHD2 and PHD3 isoforms under anaemic conditions and PHD inhibition in some tubular compartments. This effect has also been described by other research groups (D'Angelo et al., 2003;

Marxsen et al., 2004; Stiehl et al., 2006) and is likely mediated by HIF-1$\alpha$, which is predominantly expressed in tubular epithelial cells but not fibroblasts (Schley et al., 2012; Schödel et al., 2009). Our results indicate that HIF-2 is predominantly stabilized in interstitial fibroblasts during brief hypoxia or PHD inhibition, whereas sufficient HIF-1$\alpha$ can only be stabilized after PDGFR-$\beta$-specific Vhl deletion.

Our finding that EPO induction in the presence of an additional stimulus, such as anaemia, is restricted to the cortex, even in animals with PDGFR-$\beta$-specific PHD3 deletion, also argues against both compensatory PHD3 upregulation as the underlying mechanism for the lack of medullary EPO production and a role of PHD3 in general (Fig. 8). However, EPO induction was higher in the anaemic PHD3-KO mice compared to controls thereby again confirming the attenuating role of PHD3 in regulating EPO. In line with this, we and others found EPO induction almost exclusively in only PHD2 positive fibroblasts during anaemic stimulation (Fig. 9G) (Kobayashi et al., 2016). Similar results could be obtained after submaximal PHDi treatment (three serial doses of roxadustat) (Figs 8 and 9H). These findings suggest that fibroblasts with additional PHD3 expression are less susceptible to EPO production and need stronger stimuli for sufficient HIF-2 stabilization. Consistently, PHD3 has been reported to hydroxylate HIF-2$\alpha$ more efficiently than PHD2 (Appelhoff et al., 2004). However, since roxadustat non-selectively inhibits all PHD isoforms (Jatho et al., 2022), the resulting EPO expression should be independent of the cell-specific PHD expression pattern. When analysing the co-expression of EPO with PHD2 and PHD3 after maximal stimulation with roxadustat in kidneys of wild-type mice, EPO could indeed be detected in PHD2$^+$/PHD3$^+$ fibroblasts, confirming that roxadustat inhibits both PHD2 and PHD3.

Interestingly, the EPO recruitment pattern resulting from an increasing number of serial PHDi administrations mirrored the successive EPO cell recruitment observed due to increasing hypoxaemic stimuli (Eckardt et al., 1993), rather than inducing EPO production evenly throughout the cortex (Fig. 4). Based on this finding, one could assume that roxadustat is distributed unevenly in the kidney tissue – first along the cortico-medullary border, then further towards the middle cortex and finally to the cortex surface (Fig. 4). However, the analyses of HIF-2 stabilization and the induction of the HIF-2 target gene ADM rather indicate an even distribution of roxadustat in the tissue. After only three doses of roxadustat, both HIF-2 stabilization and ADM induction were evenly detected in interstitial fibroblasts across all renal zones (Fig. 5). Additional doses of roxadustat only increased the number of ADM$^+$ interstitial fibroblasts without altering their zonal distribution. The fact that HIF-2 is stabilized faster in some fibroblasts

than in others is consistent with our findings on PHD2[+] and PHD2/PHD3[+] fibroblast subpopulations. In PHD3[+] fibroblasts, HIF-2 stabilization and thus ADM induction are delayed. This probably also explains a part of the successive EPO induction from the cortico-medullary border towards the cortex surface. However, based on the even distribution of the PHD2[+] and PHD2/PHD3[+] fibroblasts, it is likely that additional factors are involved in the gradual EPO induction observed even with PHD inhibition.

However, these findings still did not explain, why EPO expression is not induced in deeper medullary fibroblasts, although HIF-2 is clearly stabilized. Different factors could influence either the activity of the HIF signalling pathway or the transcription of EPO and thus prevent EPO induction. Regarding the stability and activity of HIF, several destabilizing proteins for HIF-1$\alpha$ have already been demonstrated (Flügel et al., 2007; Seo et al., 2015). However, apart from RACK1, it is not known whether these factors have a similar effect on HIF-2$\alpha$ (Liu et al., 2007). Additionally, the transcription of HIF-2 induced genes, such as RGS4 and ADM, does not support destabilization of HIF-2$\alpha$, but rather suggests a specific EPO regulatory mechanism. Recently, a number of marker genes have been identified that are reported to be characteristic for the renal EPO cell population (Kragesteen et al., 2023). In this context, two transcription factors were also identified that may be involved in the regulation of EPO expression: Tcf21 and Cebpd (Kragesteen et al., 2023; Mao et al., 2023). Indeed, the expression of both transcription factors was regulated in interstitial fibroblasts under the conditions studied – Tcf21 expression was downregulated in fibroblasts in which HIF-2 was stabilized, while Cebpd was upregulated. However, we could not detect differences in the expression levels for either Cebpd or Tcf21 between the deeper medullary EPO-negative fibroblasts in animals after PHD inhibition and the deeper medullary EPO-positive fibroblasts in animals with genetic HIF-2 stabilization (Fig. 8). Therefore, these results do not suggest that Tcf21 or Cebpd are the key factors preventing the induction of EPO in deeper medullary fibroblasts.

In a previous study metaplastic cell transformation could be observed in connection with juxtaglomerular renin cells after renin-cell-specific HIF-2$\alpha$ stabilization (Broeker et al., 2021; Gerl et al., 2015; Kurt et al., 2015). Vhl or PHD2/PHD3 deletion led to the induction of EPO expression, whereby renin production was downregulated. In parallel, further phenotypic changes occurred, such as the expression of the marker CD73, which is typical for EPO cells, whereas typical renin cell markers such as Cx40 and Akr1b7 were downregulated. Interestingly, a single administration of a PHD inhibitor was not sufficient to induce EPO in juxtaglomerular cells, although HIF-2$\alpha$ was stabilized (Broeker et al., 2021).

In the current study, genetic HIF2 stabilization, whether by Vhl deletion or PHD2/PHD3 codeletion, did not lead to any obvious phenotypic changes in deeper medullary fibroblasts. However, it is conceivable that chronic HIF-2 stabilization might enable epigenetic changes that ultimately allow HIF-2 binding to the EPO gene locus, which does not seem to be possible with shorter periods of HIF-2 stabilization, even if HIF-2 was stabilized continuously for more than 13 h. For example, acetylation or demethylation processes could make the HIF binding site at the EPO gene accessible. Indeed, inhibition of DNA methyltransferases has previously been shown to restore EPO expression in mice with fibrotic kidney disease (Chang et al., 2016). Moreover, human EPO gene expression can also be suppressed by DNA methylation (Yin & Blanchard, 2000). In addition, the demethylase JMJD1A (Kdm3a) was identified as coactivator of EPO expression in a cell culture model (Tian et al., 2019) and might play a functional role even if its expression pattern did not differ between cortical and medullary fibroblasts. A role of HIFs in regulating epigenetic changes like DNA methylation, histone modifications or chromosomal conformational changes have been demonstrated previously (Mimura et al., 2013; Nangaku et al., 2015). Overall, it seems likely that such epigenetic changes play a role in preventing EPO induction in deeper medullary fibroblasts. Therefore, these aspects need to be analysed further in the future. To this end, we work on isolating the cortical and medullary fibroblasts separately to analyse chromatin accessibility and interaction with transcription factors in more detail.

In conclusion, our study clearly shows that HIF-2 stabilization is not sufficient to induce EPO production in deep medullary renal fibroblasts, although these cells are in principle capable of expressing EPO, as demonstrated by genetic activation of the HIF-2 pathway. Furthermore, our results demonstrate that the expression patterns of PHD2 and PHD3 in interstitial fibroblasts are not the determining factor for the restrictive cortical induction of EPO production under hypoxaemic and pharmacological conditions. However, our results confirm PHD2 as the main regulator of EPO expression in the kidney and identify PHD3 as an attenuating factor in a subset of PDGFR-$\beta^+$ cells, delaying EPO induction by raising the threshold for HIF-2 stabilization in these cells. Overall, our data suggest that additional mechanisms beyond the HIF-2 signalling pathway, such as epigenetic regulation, may be involved in controlling the EPO expression in deeper medullary fibroblasts. One could speculate that these cells provide an emergency reserve pool for EPO production that can only be activated under certain conditions, for example when cortical fibroblasts fail to produce EPO. Interestingly, in mice subjected to the kidney disease model of unilateral ureter obstruction, EPO expression

could indeed be detected in the papilla (Broeker et al., 2020; Fuchs et al., 2021). In the healthy kidney, however, these cells seem to be protected from producing EPO. Since the renal medulla is chronically hypoxic, such an additional protective mechanism certainly seems important. Otherwise, continuous EPO production in the deeper medulla would lead to chronic polycythaemia, and the well-known fine-tuned regulation of EPO in response to even small fluctuations in oxygen availability would not be possible.

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

## Additional information

### Data availability statement

All data supporting our results are included in the manuscript. Original data and micrographs are available on request. Please contact Dr Katharina Broeker (katharina.broeker@ur.de).

### Competing interests

The authors declare that they have no conflicts of interest.

### Author contributions

K.A.-E.B. conceived and designed the research studies, analysed and interpreted data and wrote the manuscript. K.A.-E.B. and B.K.-M.F. made the figures. K.A.-E.B., B.K.-M.F., M.A.A.F., L.M.S., and A.L.F. performed experiments and acquired and analysed data. B.K.-M.F., M.A.A.F., L.M.S., A.L.F., and A.K. reviewed the manuscript. A.K. interpreted data and was regularly happy to discuss the data. All authors edited the manuscript. All authors have read and approved the final version of this manuscript and agree to be accountable for all aspects of the work in ensuring that questions related to the accuracy or integrity of any part of the work are appropriately investigated and resolved. All persons designated as authors qualify for authorship, and all those who qualify for authorship are listed.

### Funding

This work was funded by the Deutsche Forschungsgemeinschaft (DFG, German Research Foundation), project number 509 149 993, TRR 374, Project B1.

### Acknowledgements

The authors would like to thank Susanne Fink for her expert technical assistance.

### Keywords

erythropoietin, hypoxia signalling, interstitial fibroblasts, PHD inhibition, prolyl-4-hydroxylases, renal endocrine function

### Supporting information

Additional supporting information can be found online in the Supporting Information section at the end of the HTML view of the article. Supporting information files available:

**Peer Review History**

