## [Peer Review History · The Journal of Physiology]

HIF-2 stabilization is not sufficient to induce EPO production in deeper medullary fibroblasts

Bettina K. M. Firmke, Michaela A.A. Fuchs, Lena Marie Süß, Anna-Lena Forst, Armin Kurtz, and Katharina Anna-Elisabeth Broeker

DOI: 10.1113/JP288798

Corresponding author(s): Katharina Broeker (katharina.broeker@ur.de)

Review Timeline:

Submission Date:	26-Feb-2025
Editorial Decision:	22-Apr-2025
Revision Received:	26-Jun-2025
Accepted:	31-Jul-2025

Senior Editor: Kim Barrett

Reviewing Editor: Morag Mansley

Transaction Report:

Dear Dr Broeker,

Re: JP-RP-2025-288798 "**HIF-2 stabilization is not sufficient to induce EPO production in deeper medullary fibroblasts**" by Bettina K. M. Firmke, Michaela A.A. Fuchs, Lena Marie Süß, Anna-Lena Forst, Armin Kurtz, and Katharina Anna-Elisabeth Broeker

Thank you for submitting your manuscript to The Journal of Physiology. It has been assessed by a Reviewing Editor and by 2 expert referees and we are pleased to tell you that it is acceptable for publication following satisfactory revision.

REVISION CHECKLIST:

We look forward to receiving your revised submission.

Yours sincerely,

Kim Barrett
Senior Editor
The Journal of Physiology

REQUIRED ITEMS

- Author photo and profile. First or joint first authors are asked to provide a short biography (no more than 100 words for one author or 150 words in total for joint first authors) and a portrait photograph. These should be uploaded and clearly labelled together in a Word document with the revised version of the manuscript. See Information for Authors for further details.

- You must start the Methods section with a paragraph headed Ethical approval (https://jp.msubmit.net/cgi-bin/main.plex?form_type=display_requirements#methods).

Research must comply with The Journal's policies regarding animal experiments (<https://physoc.onlinelibrary.wiley.com/hub/animal-experiments>) and adherence to these policies must be stated in the manuscript.

Authors should confirm in their Methods section that their experiments were carried out according to the guidelines laid down by their institution's animal welfare committee, including an ethics approval reference number. The Methods section must contain a statement about access to food, water and housing, details of the anaesthetic regime: anaesthetic used, dose and route of administration, and method of killing the experimental animals.

- Please upload separate high-quality figure files via the submission form.

- You must upload original, uncropped western blot/gel images (including controls) if they are not included in the manuscript. This is to confirm that no inappropriate, unethical or misleading image manipulation has occurred. These should be uploaded as 'Supporting information for review process only'. Please label/highlight the original gels so that we can clearly see which sections/lanes have been used in the manuscript figures. For more information, see: <https://physoc.onlinelibrary.wiley.com/hub/journal-policies#imagmanip>.

- Papers must comply with the Statistics Policy: https://jp.msubmit.net/cgi-bin/main.plex?form_type=display_requirements#statistics.

In summary:

- If $n \leq 30$, all data points must be plotted in the figure in a way that reveals their range and distribution. A bar graph with data points overlaid, a box and whisker plot or a violin plot (preferably with data points included) are acceptable formats.

- If $n > 30$, then the entire raw dataset must be made available either as supporting information, or hosted on a not-for-profit repository, e.g. FigShare, with access details provided in the manuscript.

- 'n' clearly defined (e.g. x cells from y slices in z animals) in the Methods. Authors should be mindful of pseudoreplication.

- All relevant 'n' values must be clearly stated in the main text, figures and tables.

- The most appropriate summary statistic (e.g. mean or median and standard deviation) must be used. Standard Error of the Mean (SEM) alone is not permitted.

- Exact p values must be stated. Authors must not use 'greater than' or 'less than'. Exact p values must be stated to three significant figures even when 'no statistical significance' is claimed.

- Please include an Abstract Figure file, as well as the Figure Legend text within the main article file. The Abstract Figure is a piece of artwork designed to give readers an immediate understanding of the research and should summarise the main conclusions. If possible, the image should be easily 'readable' from left to right or top to bottom. It should show the physiological relevance of the manuscript so readers can assess the importance and content of its findings. Abstract Figures should not merely recapitulate other figures in the manuscript. Please try to keep the diagram as simple as possible and without superfluous information that may distract from the main conclusion(s). Abstract Figures must be provided by authors no later than the revised manuscript stage and should be uploaded as a separate file during online submission labelled as File Type 'Abstract Figure'. Please also ensure that you include the figure legend in the main article file. All Abstract Figures should be created using BioRender. Authors should use The Journal's premium BioRender account to export high-resolution images. Details on how to use and access the premium account are included as part of this email.

EDITOR COMMENTS

Reviewing Editor:

The review of the manuscript, "HIF-2 stabilization is not sufficient to induce EPO production in deeper medullary fibroblasts" that was submitted to The Journal of Physiology is complete, having been assessed by 2 referees as well as the reviewing and senior editors.

The Editors have carefully read your manuscript and considered the points raised by the referees. Both referees believe these findings are influential and of interest to the audience of the Journal. However, both referees have raised comments that should be addressed. In addition, as a reminder - The Journal does not publish supplementary material, therefore any supplementary figures which are essential should be incorporated into the current figures and referred to within the text. If not, these should be removed.

REFEREE COMMENTS

Referee #1:

In their manuscript titled: "HIF-2 stabilization is not sufficient to induce EPO production in deeper medullary fibroblasts" Firmke and colleagues investigate mechanisms that regulate kidney EPO production in mice exposed to different EPO-inducing stimuli. The investigators use genetic models, an anemic hypoxia model (moderate), and pharmacological means to induce HIF-2 dependent EPO expression in EPO-competent kidney interstitial fibroblast/pericytes and identify a subpopulation of EPO-competent medullary fibroblasts that respond with HIF2a stabilization but fail to produce EPO under anemic conditions and in response to HIF-PHI treatment.

The studies expand previous investigations from this group and advance the field.

Although mostly descriptive, the data are likely to stimulate further investigations in this area and, in the opinion of this reviewer, are well-suited for publication in this journal.

To improve the impact of the manuscript, this reviewer has a few suggestions. These are not major modifications and can be accomplished in a timely manner.

1. It would be helpful, if the authors provided a table summarizing the data, i.e., conditions and zones that associate with HIF-2a, EPO and other HIF regulated genes. e.g., HIF2a + , EPO -, etc. - the table could include data published previously.

2. The authors may want to comment on the outer and inner stripe of the outer medulla and may draw an additional demarcation line.
3. Please explain what the authors mean by emergency response when there is already a lot of EPO being produced by the kidney cortex. Is the additional medullary production of EPO really going to increase the effect on the bone marrow or are their other effects that are being referred to.
4. It would be helpful to add some tubular markers to the FISH studies, FISH-IF, e.g., especially in the context of Phd3 FISH. These images are difficult to interpret. In the result sections please state your findings first, e.g., co-localization of signals, and not just your interpretation of data.
5. Please add arrows to the images to indicate e.g., which cells co-express Phd3 and other genes.
6. Please state in the figure legends and/or main text which x admin of roxa was used for the HIF-PHI condition; always 8x ? This is not clearly stated. Address the tissue distribution of roxa in the discussion.

Referee #2:

In the manuscript from Firmke et al., the authors investigate the important and relevant question how the recruitment of Epo-producing cells is physiologically regulated or prevented, respectively. It is still unclear why many cells exist throughout the kidney that have the capability to produce Epo but are not recruited during e. g. anemia. Here, the authors provide evidence that the main transcription factor regulating Epo expression in the kidney, HIF-2, can be activated by a PHDi even in the medulla. However, Epo expression is not induced despite HIF-2 activation. This indicates that there are regulations in place that determine whether a cell expresses Epo or not that are beyond the HIF-2 pathway. The results and descriptions of the authors are convincing, and I only have a few comments.

- The authors used the PHDi Roxadustat for their investigation. There are 5 additional PHDi's approved for treatment of renal anemia (globally). Can the authors comment on whether it is known if some or all of these inhibitors lead to the same renal Epo expression pattern and HIF-2 activation as Roxadustat or whether there is a difference based on the used inhibitor? Could it be that the described differences between hypoxemia/PHDi treatment and VHL deletion are observed, because induced hypoxemia does not reach deep enough into the medulla and the inhibitor effect is specific to Roxadustat?
- Does the interstitial PHD2 and PHD3 expression level change in between the different kidney areas (cortex, outer and inner medulla)?
- p. 13, lines 346-348 and p. 16, 374-375: In both statements, it is unclear whether these results are shown in the manuscript or not. Please, either indicate in which figure(s) these results can be found or include the corresponding results.
- It would aid the understanding and interpretation of the results, when the amount of the used PHDi together with the treatment time/repetition is mentioned in the figure legends.
- Fig. 2: Please, highlight (at least some) positive cells.

END OF COMMENTS

Dear Professor Barrett,

Dear reviewers,

Thank you all for your helpful and much-appreciated comments.

You will find our responses below. We hope we have addressed all concerns satisfactorily.

On behalf of all the authors,

Yours sincerely,

Katharina Broecker

EDITOR COMMENTS

Reviewing Editor: The review of the manuscript, "HIF-2 stabilization is not sufficient to induce EPO production in deeper medullary fibroblasts" that was submitted to The Journal of Physiology is complete, having been assessed by 2 referees as well as the reviewing and senior editors.

The Editors have carefully read your manuscript and considered the points raised by the referees. Both referees believe these findings are influential and of interest to the audience of the Journal. However, both referees have raised comments that should be addressed. In addition, as a reminder - The Journal does not publish supplementary material, therefore any supplementary figures which are essential should be incorporated into the current figures and referred to within the text. If not, these should be removed.

We have either incorporated previously submitted supplementary figures into the main figures of the manuscript or added them as new figures if integration into an existing figure did not seem appropriate.

REFEREE COMMENTS

Referee #1:

In their manuscript titled: "HIF-2 stabilization is not sufficient to induce EPO production in deeper medullary fibroblasts" Firmke and colleagues investigate mechanisms that regulate kidney EPO production in mice exposed to different EPO-inducing stimuli. The investigators use genetic models, an anemic hypoxia model (moderate), and pharmacological means to induce HIF-2 dependent EPO expression in EPO-competent kidney interstitial fibroblast/pericytes and identify a subpopulation of EPO-competent medullary fibroblasts that respond with HIF2a stabilization but fail to produce EPO under anemic conditions and in response to HIF-PHI treatment.

The studies expand previous investigations from this group and advance the field.

Although mostly descriptive, the data are likely to stimulate further investigations in this area and, in the opinion of this reviewer, are well-suited for publication in this journal.

To improve the impact of the manuscript, this reviewer has a few suggestions. These are not major modifications and can be accomplished in a timely manner.

1. It would be helpful, if the authors provided a table summarizing the data, i.e., conditions and zones that associate with HIF-2a, EPO and other HIF regulated genes. e.g., HIF2a +, EPO -, etc. - the table could include data published previously.

As suggested, we added a table (table 4) that summarizes the expression profile of interstitial fibroblasts of the different renal zones under different conditions. As suggested, we considered including data that we had already published in previous studies. However, we decided against it, as we feel that the inclusion of markers such as CD73 or Tenascin-C could confuse the reader, given that they are not mentioned elsewhere in this study.

2. The authors may want to comment on the outer and inner stripe of the outer medulla and may draw an additional demarcation line.

We consider the outer stripe of the outer medulla to be the area of the outer medulla in which the S3 segments of the proximal tubules are located. The inner stripe of the outer medulla is the part of the outer medulla between the outer zone of the outer medulla and the inner medulla (= papilla). The inner stripe of the outer medulla is mostly constituted of the thin descending and thick ascending limbs of the loops of Henle, as well as collecting ducts. We included a respective statement in the manuscript text (results section). We also added additional demarcation lines in the figures (Fig. 4, Fig. 5 and Fig. 8).

3. Please explain what the authors mean by emergency response when there is already a lot of EPO being produced by the kidney cortex. Is the additional medullary production of EPO really going to increase the effect on the bone marrow or are there other effects that are being referred to.

The term “emergency reserve pool” may have been somewhat misleading. We believe that in the healthy kidney under physiological conditions further regulatory mechanisms (e.g. epigenetic mechanisms) ensure that no EPO is induced in deeper medullary fibroblasts even when HIF-2 is stabilized. Accordingly, we do not think that EPO induction in these cells could have an additional effect on the bone marrow, when the cortical and outer medullary fibroblasts also produce a lot of EPO in parallel due to a hypoxic stimulus/PHD inhibition. However, we had observed in two previous studies that in the renal disease model of unilateral ureteral occlusion (UUO), where there is a reduction in EPO production in the cortical fibroblasts, EPO was surprisingly induced in the papilla (Broeker *et al.*, 2020; Fuchs *et al.*, 2021). Furthermore, in this and previous studies we could show that with genetic, i.e. chronic stabilization of HIF-2, the fibroblasts of all kidney zones were capable of EPO production (regardless of whether PHD2, PHD2/PHD3 or Vhl was deleted specifically in fibroblasts). Thus, fibroblasts of the deeper medullary zones could be special reserve cells for EPO production, which under physiological (normoxia, hypoxia/anemia, PHD inhibition <15 hours) conditions are protected from EPO induction by additional regulatory mechanisms, but seem to be able to start EPO production in very specific situations/conditions, such as fibrotic kidney diseases (UUO model), when EPO production in the “classic” EPO-producing fibroblasts is restricted.

To avoid any misunderstandings, this sentence has been rephrased in both the 'Key Points' section and the 'Abstract'.

4. It would be helpful to add some tubular markers to the FISH studies, FISH-IF, e.g., especially in the context of Phd3 FISH. These images are difficult to interpret. In the result sections please state your findings first, e.g., co-localization of signals, and not just your interpretation of data.

As suggested, we performed new RNAscope assays that include the tubular marker cadherin16 to make it easier to interpret coexpression of PHD2 or PHD3 with PDGFR- β or coexpression of PHD2 or PHD3 with tubular structures (Fig. 9). Interstitial coexpression was highlighted with arrows. Increased expression of either PHD2 or PHD3 in tubules is indicated with asterisks. Increased expression of PHD3 in PDGFR- β^+ or EPO $^+$ fibroblasts is indicated with arrowheads. We have also tried to clearly state our findings before providing interpretations.

5. Please add arrows to the images to indicate e.g., which cells co-express Phd3 and other genes.

Where applicable, we added arrows to the images to highlight coexpression of genes.

6. Please state in the figure legends and/or main text which x admin of roxa was used for the HIF-PHI condition; always 8x? This is not clearly stated.

We apologize for any confusion. We have added information to the figure legends and/or the text to clearly state how many roxadustat applications were used in the respective experiment/treatment group.

Address the tissue distribution of roxa in the discussion.

We have added a paragraph to the discussion in which we discuss the distribution of roxadustat in the tissue.

Examining the induction of EPO resulting from an increasing number of serial administrations of roxadustat revealed the following EPO expression patterns: Following a single administration of roxadustat, EPO induction was initially detectable in interstitial fibroblasts along the cortico-medullary border. After three administrations, additional fibroblasts were recruited for EPO production, predominantly in the outer stripe of the outer medulla and the deep and middle cortex. After six administrations of roxadustat, EPO induction extended to the cortical surface. After eight administrations, the zonal distribution of EPO induction remained unchanged; only the number of EPO-producing cells per kidney section increased slightly. Based on this, one could initially assume that roxadustat is distributed unevenly in the kidney tissue - first along the cortico-medullary border, then further towards the middle cortex and finally to the cortex surface. However, analysis of HIF-2 stabilization on kidney sections, as well as the induction of other HIF target genes, such as ADM, showed different results. HIF-2 stabilization could already be detected in interstitial fibroblasts throughout the cortex, outer medulla, and inner medulla after three administrations of roxadustat. A similar pattern emerged for ADM induction. ADM induction in PDGFR- β^+ fibroblasts was evenly distributed across all kidney zones (see Fig. 5). Additional doses of roxadustat increased the number of ADM $^+$ interstitial fibroblasts in all renal zones without altering their zonal distribution (see Fig. 3). This indicates that roxadustat is distributed evenly throughout renal tissue.

Referee #2:

In the manuscript from Firmke et al., the authors investigate the important and relevant question how the recruitment of Epo-producing cells is physiologically regulated or prevented, respectively. It is still unclear why many cells exist throughout the kidney that have the capability to produce Epo but are not recruited during e. g. anemia. Here, the authors provide evidence that the main transcription factor regulating Epo expression in the kidney, HIF-2, can be activated by a PHDi even in the medulla. However, Epo expression is not induced despite HIF-2 activation. This indicates that there are regulations in place that determine whether a cell expresses Epo or not that are beyond the HIF-2 pathway. The results and descriptions of the authors are convincing, and I only have a few comments.

- The authors used the PHDi Roxadustat for their investigation. There are 5 additional PHDi's approved for treatment of renal anemia (globally). Can the authors comment on whether it is known if some or all of these inhibitors lead to the same renal Epo expression pattern and HIF-2 activation as Roxadustat or whether there is a difference based on the used inhibitor? Could it be that the described differences between hypoxemia/PHDi treatment and VHL deletion are observed, because induced hypoxemia does not reach deep enough into the medulla and the inhibitor effect is specific to Roxadustat?

Nakai and colleagues compared efficiency of the approved PHDi's – daprodustat, enarodustat, molidustat, vadadustat and roxadustat – by analyzing renal EPO mRNA induction levels as well as resulting plasma EPO concentrations in C57BL/6 mice. Mice were injected intraperitoneally with a single dose of the respective PHDi (200mg/kg body weight) 6 hours prior to analysis. They reported no significant differences between the different PHDi's in terms of EPO mRNA abundance in the kidney or EPO concentration in the plasma (Nakai *et al.*, 2022, 2024). Unfortunately, they did not analyze the stabilization of HIF-2 or the tissue distribution of EPO.

Kobayashi and colleagues used molidustat to analyze the effect of PHD inhibition in the context of a chronic kidney disease model. The control mice in the study were treated orally with a single dose of molidustat (10mg/kg) 4 hours prior to analysis. EPO induction was detected by RNAscope predominantly in the outer stripe of the outer medulla and inner and midcortex. Moreover, they reported serum EPO concentrations of about 7000-8000 pg/ml (Kobayashi *et al.*, 2022). Regarding EPO induction, these findings are consistent with our results from treating mice with roxadustat three times at 90-minute intervals and analyzing the kidneys and plasma 90 minutes after the last roxadustat administration (i.e., 4.5 hours after the initial administration) (please see Fig. 4 and 8). Unfortunately, they did not investigate HIF-2 stabilization, or the induction pattern of other HIF-2 target genes.

We could not find any reports on the tissue distribution of EPO induction, HIF-2 stabilization or the tissue induction pattern of other HIF-2 target genes for daprodustat, enarodustat or vadadustat.

However, in a previous study, some colleagues and I treated mice with the PHD inhibitor ICA (2-(1-Chloro-4-hydroxyisoquinoline-3-carboxamido)acetate; not approved for patients) to analyze EPO induction in the context of kidney disease. EPO induction was also analyzed in healthy control kidneys 4 hours after injection of a single dose of the PHD inhibitor ICA (i.p.; 40 mg/kg body weight). Similarly to roxadustat and molidustat, EPO was induced only in the outer stripe of the outer medulla and throughout the cortex (Fuchs *et al.*, 2021). Again, HIF-2 stabilization or induction of other HIF-2 target genes was not investigated.

Since we still had tissue from the mentioned study, we performed a co-RNAscope for ADM, EPO and PDGFR- β . As described, EPO induction was limited to the cortex and the outer

stripe of the outer medulla. However, ADM induction extended across all renal zones into the papilla, similar to 3x roxadustat (see Fig. 1, below).

Figure 1: Recruitment pattern of renal ADM and EPO expression on kidney sections of wildtype mice treated with the PHDi ICA.

A: Distribution of ADM/PDGFR- β -expressing cells (yellow dots) and ADM/EPO/PDGFR- β -expressing cells (pink dots) on kidney sections of wildtype mice treated with ICA. ADM/PDGFR- β coexpression was detected across all kidney zones, whereas coexpression of ADM/EPO/PDGFR- β could only be observed in the outer stripe of the outer medulla and throughout the cortex. Dotted white lines indicate the zonal borders. Nuclei were counterstained with Dapi (grey). Scale bars: 500 μ m.

B/C/D/E: Details showing the colocalization of ADM (red) with PDGFR- β (green) and EPO (blue) across all kidney zones (cortex, outer medulla = OM, inner medulla =IM) using RNAscope. ADM could be detected in PDGFR- β^+ fibroblasts across all kidney zones. The EPO-producing cells in the cortex and outer stripe of the outer medulla were also positive for ADM and PDGFR- β . Arrows highlight exemplarily some ADM-expressing (EPO $^+$) fibroblasts. Nuclei were counterstained with Dapi (grey). Scale bars: 20 μ m.

Regarding HIF-2 stabilization Rosenberger and colleagues did an extensive study on hypoxic rat kidneys. Hypoxia was either induced by phlebotomy (rats were anemic with hematocrit values of 16%; euthanized 3 hours later) or by exposing the rats to 0.1% CO for 5 hours. The authors reported similar staining patterns for HIF-2 α under both conditions. HIF-2 stabilization could be detected in interstitial cells of all kidney zones (cortex, outer medulla (outer and inner stripe) and inner medulla) except in the papilla. Additionally, endothelial cells and some intraglomerular cells were positive for HIF-2 α (Rosenberger *et al.*, 2002). This reported HIF-2 stabilization pattern is very similar to the pattern, we observed after 8x roxadustat.

Despite these findings, EPO induction in both rats and mice under various hypoxic stimuli (0.1% CO, low O₂ or anemia) is only ever reported in the outer stripe of the outer medulla and in the cortex (Bachmann *et al.*, 1993; Eckardt *et al.*, 1993; Suzuki *et al.*, 2016; Broeker *et al.*, 2020). Interestingly, Sandner and colleagues reported expression of ADM in interstitial cells of the inner medulla in rat kidneys after CO-exposure (Sandner *et al.*, 2004).

Taken together, we agree with this reviewer that anemia with hematocrit values of about 25% was probably not sufficient to stabilize HIF-2 in the deeper medullary regions. However, based on the reported results of the above studies, we would conclude that our observations are not due to a specific inhibitor effect of roxadustat, but that HIF-2 stabilization - whether by

PHD inhibition, sufficiently severe anemia $\leq 16\%$ or CO exposure - is not sufficient to stabilize EPO in the inner stripe of the outer medulla or the inner medulla.

- Does the interstitial PHD2 and PHD3 expression level change in between the different kidney areas (cortex, outer and inner medulla)?

We added new data showing the interstitial expression level of PHD2 and PHD3 in the different kidney areas (Fig. 9J/K). Automated RNAscope analysis using the Zeiss Intellesis software showed that in each kidney zone about 90% of interstitial PDGFR- β^+ cells coexpressed PHD2 (Fig. 9J). In contrast, PHD3 expression could only be detected in about 35% of interstitial PDGFR- β^+ in each kidney zone by automated PHD3/PDGFR- β coexpression analysis (Fig. 9J).

To assess the cellular expression levels of PHD2 or PHD3 in interstitial fibroblasts the number of PHD2- or PHD3- positive signal dots per PDGFR- β^+ cell was scored. More than 90% of interstitial PHD2/PDGFR- β coexpressing cells showed only 1-5 PHD2 signals per cell (score 1) in the cortex and outer medulla, while about 7.5% of the cells had a score of 2 (6-10 RNAscope signal dots). In the inner medulla, over 98% of the PHD2/PDGFR- β^+ cells were assigned a score of 1 (Fig. 9K). The PHD3 expression level per interstitial PDGFR- β^+ cell was also quite similar across renal zones. In the cortex and outer medulla about 95% of interstitial PHD3/PDGFR- β coexpressing cells were assigned a score of 1, while about 4-5% of the cells had a score of 2. In the inner medulla, about 98% of the PHD3/PDGFR- β^+ cells showed a score of 1 (Fig. 9K).

Overall, there is no apparent difference in the interstitial expression level between kidney zones for either PHD2 or PHD3.

- p. 13, lines 346-348 and p. 16, 374-375: In both statements, it is unclear whether these results are shown in the manuscript or not. Please, either indicate in which figure(s) these results can be found or include the corresponding results.

We added a new figure showing HIF-2 stabilization and induction of ADM after the serial application of three doses of roxadustat (Fig. 5). Three doses of roxadustat were sufficient to stabilize HIF-2 in the deeper medullary regions. In parallel, ADM was induced in interstitial fibroblasts from all renal zones.

- It would aid the understanding and interpretation of the results, when the amount of the used PHDi together with the treatment time/repetition is mentioned in the figure legends.

We have updated the figure legends and/or text to clearly indicate the number of roxadustat applications used in each experiment.

- Fig. 2: Please, highlight (at least some) positive cells.

We have highlighted exemplarily some positive cells in each detail using arrows.

References

- Bachmann S, Le Hir M & Eckardt KU (1993). Co-localization of erythropoietin mRNA and ecto-5'-nucleotidase immunoreactivity in peritubular cells of rat renal cortex indicates that fibroblasts produce erythropoietin. *J Histochem Cytochem Off J Histochem Soc* **41**, 335–341.
- Broeker KAE, Fuchs MAA, Schrankl J, Kurt B, Nolan KA, Wenger RH, Kramann R, Wagner C & Kurtz A (2020). Different subpopulations of kidney interstitial cells produce erythropoietin and factors supporting tissue oxygenation in response to hypoxia in vivo. *Kidney Int* **98**, 918–931.
- Eckardt K-U, Koury ST, Tan CC, Schuster SJ, Kaissling B, Ratcliffe PJ & Kurtz A (1993). Distribution of erythropoietin producing cells in rat kidneys during hypoxic hypoxia. *Kidney Int* **43**, 815–823.
- Fuchs MAA, Broeker KAE, Schrankl J, Burzlaff N, Willam C, Wagner C & Kurtz A (2021). Inhibition of transforming growth factor β 1 signaling in resident interstitial cells attenuates profibrotic gene expression and preserves erythropoietin production during experimental kidney fibrosis in mice. *Kidney Int* **100**, 122–137.
- Kobayashi H, Davidoff O, Pujari-Palmer S, Drevin M & Haase VH (2022). EPO synthesis induced by HIF-PHD inhibition is dependent on myofibroblast transdifferentiation and colocalizes with non-injured nephron segments in murine kidney fibrosis. *Acta Physiol* **235**, e13826.
- Nakai T, Saigusa D, Iwamura Y, Matsumoto Y, Umeda K, Kato K, Yamaki H, Tomioka Y, Hirano I, Koshiba S, Yamamoto M & Suzuki N (2022). Esterification promotes the intracellular accumulation of roxadustat, an activator of hypoxia-inducible factors, to extend its effective duration. *Biochem Pharmacol* **197**, 114939.
- Nakai T, Saigusa D, Kato K, Fukuuchi T, Koshiba S, Yamamoto M & Suzuki N (2024). The drug-specific properties of hypoxia-inducible factor-prolyl hydroxylase inhibitors in mice reveal a significant contribution of the kidney compared to the liver to erythropoietin induction. *Life Sci* **346**, 122641.
- Rosenberger C, Mandriota S, Jürgensen JS, Wiesener MS, Hörstrup JH, Frei U, Ratcliffe PJ, Maxwell PH, Bachmann S & Eckardt K-U (2002). Expression of Hypoxia-Inducible Factor-1 α and -2 α in Hypoxic and Ischemic Rat Kidneys. *J Am Soc Nephrol* **13**, 1721–1732.
- Sandner P, Hofbauer KH, Tinel H, Kurtz A, Thiesson HC, Ottosen PD, Walter S, Skøtt O & Jensen BL (2004). Expression of adrenomedullin in hypoxic and ischemic rat kidneys and human kidneys with arterial stenosis. *Am J Physiol Regul Integr Comp Physiol* **286**, R942-951.
- Suzuki N, Sasaki Y, Kato K, Yamazaki S, Kurasawa M, Yorozu K, Shimonaka Y & Yamamoto M (2016). Efficacy estimation of erythropoiesis-stimulating agents using erythropoietin-deficient anemic mice. *Haematologica* **101**, e356-360.

Dear Dr Broeker,

Re: JP-RP-2025-288798R1 "**HIF-2 stabilization is not sufficient to induce EPO production in deeper medullary fibroblasts**" by Bettina K. M. Firmke, Michaela A.A. Fuchs, Lena Marie Süß, Anna-Lena Forst, Armin Kurtz, and Katharina Anna-Elisabeth Broeker

We are pleased to tell you that your paper has been accepted for publication in The Journal of Physiology.

Yours sincerely,

Kim Barrett
Senior Editor
The Journal of Physiology

If you would like to receive our 'Research Roundup', a monthly newsletter highlighting the cutting-edge research published in The Physiological Society's family of journals (The Journal of Physiology, Experimental Physiology, Physiological Reports, The Journal of Nutritional Physiology and The Journal of Precision Medicine: Health and Disease), please click this link, fill in your name and email address and select 'Research Roundup':
<https://www.physoc.org/journals-and-media/membernews>

- You can help your research get the attention it deserves! Check out Wiley's free Promotion Guide for best-practice recommendations for promoting your work at: www.wileyauthors.com/eeo/guide. You can learn more about Wiley Editing Services which offers professional video, design, and writing services to create shareable video abstracts, infographics, conference posters, lay summaries, and research news stories for your research at: www.wileyauthors.com/eeo/promotion.

EDITOR COMMENTS

Reviewing Editor:

We thank the authors for taking the time to carefully modify the manuscript based on the comments given following the initial reviews. All comments have been addressed and these changes have brought together a very nice study containing

findings of influence and impact in the field.

REFEREE COMMENTS

Referee #1:

I have no additional comments and all of my concerns have been addressed.

Referee #2:

The authors answered my comments sufficiently and at great detail.